

# The emission of CO from tropical rain forest soils

Hella van Asperen[1,2], Thorsten Warneke[1], Alessandro Carioca de Araújo[3,4], Bruce Forsberg[3], Sávio José Filgueiras Ferreira[5], Thomas Röckmann[6], Carina van der Veen[6], Sipko Bulthuis[7], Leonardo Ramos de Oliveira[3], Thiago de Lima Xavier[3], Jailson da Mata[3], Marta de Oliveira Sá[8], Paulo Ricardo Teixeira[3], Julie Andrews de França e Silva[3], Susan Trumbore[2], and Justus Notholt[1]

[1]Institute of Environmental Physics (IUP), University of Bremen, Otto-Hahn-Allee 1, Bremen, 28359, Germany
[2]Max Planck Institute for Biogeochemistry (MPI-BGC), Hans-Knöll-Strasse 10, Jena, 07745, Germany
[3]Programa de Grande Escala da Biosfera-Atmosfera na Amazônia (LBA), Instituto Nacional de Pesquisas da Amazônia (INPA), Av. André Araújo, 2936, Petrópolis, AM 69067-375, Manaus, Brazil
[4]Brazilian Agricultural Research Corporation (EMBRAPA), Embrapa Amazônia Oriental, Tv. Dr. Enéas Piheiro, s/n, Marco, PA 66095-903, Caixa postal 48, Belém, Brazil
[5]Coordenação de Dinâmica Ambiental (CODAM), Instituto Nacional de Pesquisas da Amazônia (INPA), Av. André Araújo, 2936, Petrópolis, AM 69067-375, Manaus, Brazil
[6]Institute for Marine and Atmospheric Research Utrecht, Utrecht University, Princetonplein 5, 3584, Utrecht, Netherlands
[7]Max Planck Institute for Chemistry, Hahn-Meitner-Weg 1, Mainz, 55128, Germany
[8]PhD student of Postgraduate Program in Climate and Environment (CLIAMB) - INPA/UEA, Instituto Nacional de Pesquisas da Amazônia (INPA), Av. André Araújo, 2936, Petrópolis, AM 69067-375, Manaus, Brazil

**Correspondence:** Hella van Asperen (hasperen@bgc-jena.mpg.de)

**Abstract.** Soil CO fluxes represent a net balance between biological soil CO uptake and abiotic soil and (senescent) plant CO production. Studies largely from temperate and boreal forests indicate that soils serve as a net sink for CO, but to date uncertainty remains about the role of tropical rain forest soils. Here we report the first direct measurements of soil CO fluxes in a tropical rain forest. We compare with estimates of net ecosystem CO fluxes derived from accumulation of CO at night under stable atmospheric conditions. Further, we used laboratory experiments to demonstrate the importance of temperature on net soil CO fluxes. Net soil surface CO fluxes ranged from -0.19 to 3.36 nmol m$^{-2}$ s$^{-1}$, averaging ∼1 nmol CO m$^{-2}$ s$^{-1}$. Fluxes varied with season and topographic location, with highest fluxes measured in the dry season in a seasonally inundated valley. Ecosystem CO fluxes estimated from nocturnal canopy air profiles, which showed CO mixing ratios that consistently decreased with height, ranged between 0.3 and 2.0 nmol CO m$^{-2}$ s$^{-1}$. A canopy layer budget method, using the nocturnal increase in CO, estimated similar flux magnitudes (1.1 to 2.3 nmol CO m$^{-2}$ s$^{-1}$). In the wet season, a greater valley ecosystem CO production was observed in comparison to measured soil valley CO fluxes, suggesting a contribution of the valley stream to overall CO emissions. Laboratory incubations demonstrated a clear increase in CO production with temperature that was also observed in field fluxes, though high correlations between soil temperature and moisture limit our ability to interpret the field relationship. At a common temperature (25 °C), expected plateau and valley senescent leaf CO production was small (0.012 and 0.002 nmol CO m$^{-2}$ s$^{-1}$) in comparison to expected soil material CO emission (∼0.9 nmol CO m$^{-2}$ s$^{-1}$). Based on our field and laboratory observations, we expect that tropical rain forest ecosystems are a net source of CO, with thermal degradation-induced soil emissions likely being the main contributor to ecosystem CO emissions. Extrapolating our first observation-based tropical rain forest soil emission estimate of ∼1 nmol m$^{-2}$ s$^{-1}$, a global tropical rain forest soil emission of





$\sim$16.0 Tg CO yr$^{-1}$ is estimated. Nevertheless, total ecosystem CO emissions might be higher, since valley streams and inun-
dated areas might represent local CO emission hot spots. To further improve tropical forest ecosystem CO emission estimates,
more in-situ tropical forest soil and ecosystem CO flux measurements are essential.

## 1 Introduction

Carbon monoxide (CO) is a trace gas in the atmosphere. It is the most important sink for the hydroxyl (OH) radical, which
also serves as a sink for methane (CH$_4$). Thus, an increase in CO emissions will directly affect the atmospheric concentrations
of CH$_4$, making CO an indirect greenhouse gas, with a possible indirect radiative forcing larger than N$_2$O (Szopa et al., 2021).
Anthropogenic activities, such as (incomplete) combustion of fossil fuel and biomass, contribute strongly to global CO emis-
sions, and CO concentrations in urban areas are usually higher than in rural areas (Zheng et al., 2019; Seinfeld and Pandis,
2016). Due to its short atmospheric lifetime of 50 days, spatial differences between regions can be large and concentrations in
the northern hemisphere are generally higher than in the southern hemisphere (Szopa et al., 2021; Seinfeld and Pandis, 2016).
Besides direct anthropogenic emissions, CO is also produced by atmospheric oxidation sources, such as the in-situ oxidation of
methane and hydrocarbons, or can be emitted by (partly) natural sources such as forest fires, ocean emissions, the degradation
of chlorophyll, and abiotic degradation of organic matter (Sanderson, 2002; Szopa et al., 2021; Seinfeld and Pandis, 2016).
The major natural sinks of carbon monoxide are the tropospheric oxidation with OH ($>$80%), the uptake by soils ($\sim$10-15%),
and the removal in the stratosphere ($\sim$5%) (Seinfeld and Pandis, 2016; King and Weber, 2007; Bartholomew and Alexander,
1979; Conrad, 1996; Sanderson, 2002; Khalil and Rasmussen, 1990).

On ecosystem level, the sources and sinks of CO are poorly understood. Soils can act as net sources as well as sinks of
CO (Conrad, 1996). Most likely, the main process involved in soil CO uptake is the oxidation of CO to CO$_2$, or the reduc-
tion to CH$_4$ by soil bacteria or soil enzymes (Bartholomew and Alexander, 1979; Conrad, 1996; Ingersoll et al., 1974; Spratt
and Hubbard, 1981; Whalen and Reeburgh, 2001; Yonemura et al., 2000; Ingersoll et al., 1974). Soil CO consumption was re-
ported to be poorly related to temperature (Conrad and Seiler, 1985), and more related to soil diffusivity (Sanhueza et al., 1994;
Conrad and Seiler, 1982; Kisselle et al., 2002). Soil CO emissions are thought to be mostly of non-biological origin, namely
photo-degradation (Bruhn et al., 2013; Pihlatie et al., 2016; Schade et al., 1999; Lee et al., 2012; Tarr et al., 1995; Derendorp
et al., 2011) and thermal degradation (Conrad and Seiler, 1980; Derendorp et al., 2011; van Asperen et al., 2015; Conrad and
Seiler, 1982; Lee et al., 2012; Yonemura et al., 2000). Besides emissions associated with abiotic degradation of organic matter,
living plants are also known to emit small amounts of CO (Bruhn et al., 2013; Kirchhoff and Marinho, 1990; Tarr et al., 1995).
However, emissions from senescent plant material are 5 to 10 times greater than those observed from photosynthesising leaf
material (Derendorp et al., 2011; Schade et al., 1999; Tarr et al., 1995).



Soil CO fluxes thus represent the net balance between biological soil CO uptake and abiotic soil and (senescent) plant CO production (Whalen and Reeburgh, 2001; Liu et al., 2018; Potter et al., 1996; van Asperen et al., 2015; Pihlatie et al., 2016; Constant et al., 2008). Besides temperature and radiation, it has been observed that the net flux is dependent on, among others, soil water content, soil organic carbon, land use type, and nutrients (King, 2000; King and Hungria, 2002; Conrad and Seiler,
1985; Funk et al., 1994; Gödde et al., 2000; Yonemura et al., 2000; Moxley and Smith, 1998). Due to its dependency on environmental factors, the net CO flux balance might shift diurnally and seasonally. Existing measurements of diurnal cycles mostly show a shift towards uptake during nighttime hours, and emission during daytime hours (van Asperen et al., 2015; Sanhueza et al., 1994; Scharffe et al., 1990; Schade et al., 1999). The few long term CO flux studies found a similar pattern seasonally, with increased uptake during colder periods, and more emission during warmer periods (Constant et al. (2008);
Cowan et al. (2018); Pihlatie et al. (2016)).

Previous CO flux measurements have been done in boreal ecosystems (Constant et al., 2008; Laasonen et al., 2021; Pihlatie et al., 2016; Whalen and Reeburgh, 2001; Funk et al., 1994), temperate zones (Cowan et al., 2018; Conrad et al., 1988; Gödde et al., 2000), and arid and (sub-)tropical ecosystems (Sanhueza et al., 1994; Kisselle et al., 2002; King and Hungria, 2002;
King, 2000; Scharffe et al., 1990; van Asperen et al., 2015), but we are aware of no previous CO flux measurements from tropical rain forests. Because of this, the net CO flux of tropical forest soils predicted using global models remains highly uncertain even as to sign: while Potter et al. (1996) modelled that tropical soils are likely a source of CO, thereby implying that abiotic emission dominates over soil biological CO uptake, a more recent modelling study suggested that tropical soils are possibly a net sink of CO (Liu et al., 2018). This discrepancy shows the need for in-situ observation of soil and ecosystem CO
fluxes in tropical rain forests.

In this study, we present results from 2 intensive measurement campaigns in a tropical rain forest in central Amazon. During wet and dry season campaigns, CO fluxes were estimated in two ways. First, soil chambers enclosing both litter and soil were used to measure net surface CO and $CO_2$ fluxes. Second, above and below canopy CO and $CO_2$ mixing ratio patterns were
studied to estimate ecosystem CO fluxes from the net change of gases during stable atmospheric nocturnal conditions when mixing with air above the canopy is limited. Both methods demonstrated that tropical rain forests are a net source of CO. Third, using a simple laboratory experiment, we show that soils are the main source driving these emissions and that abiotic thermal degradation is likely its main driver. Finally by focusing on different seasons and topographic locations, we attempt to identify the role of additional CO sources in the ecosystem. Based on our observations, we formulate a first observation-based
estimate for global tropical rain forest soil CO emissions.





## 2    Material and methods

### 2.1    Field site and K34 micro-meteorological measurements

This research was performed in a mature rain forest, located ∼50 km northwest of Manaus (Brazil) at the Reserva Biológica do
Cuieiras (2 °36" 32.67 S, 60 °12"33.48 W), managed by the Instituto Nacional de Pesquisas da Amazônia (INPA), also known
as ZF2. The elevation at the site ranges from 40-110 m above sea level and is characterized by a dissected topography with
plateaus, steep slopes and valleys. The vegetation on plateaus is terra firma (upland) forest with tree heights of 35-40 m, and
with clay rich soils classified as Oxisols and Ultisols. Valleys are periodically inundated, with three heights of 25-30 m, and
with sandy soils, classified as Spodosols (Luizão et al., 2004; Zanchi et al., 2014). The field site has a distinct seasonality, with
a dry season (months with precipitation <100 mm) lasting ∼3 months between June-October, and a wet season from December
to May. Annual average precipitation is 2400 mm, and average annual air temperature is 26-28 °C. More information about
the field site can be found in Araújo et al. (2002); Chambers et al. (2004); Luizão et al. (2004); Quesada et al. (2010); Zanchi
et al. (2014).

The K34 tower is a micro-meteorological tower located at field site ZF2, run by the project LBA (Large Biosphere Atmo-
sphere Experiment) since 1999, and is one of the longest running flux towers in a tropical rain forest. The tower is equipped
with micro-meteorological as well as environmental measurements. Unfortunately, due to pandemic challenges, no measure-
ments are available for the campaign periods, but data from earlier years was available to support our analyses.

### 2.2    Available instruments: FTIR-analyzer & ICOS-Analyzer

At the foot of the K34 tower, a Fourier Transform Infrared Spectrometer (ACOEM Spectronus, Trace Greenhouse Gas and
Isotope Analyzer, from here on called FTIR-analyzer, Griffith et al. (2012)) was installed in an air-conditioned cabin. The
FTIR-analyzer simultaneously measures mixing ratios of $CO_2$, $CH_4$, $N_2O$ and CO, as well as the $\delta^{13}C$ of $CO_2$. The instru-
ment can measure in either static or flow modes. All incoming air samples are internally dried by a Nafion dryer and by a
column of magnesium perchlorate, so that $H_2O$ mixing ratios are usually <20 ppm. Measurements were corrected for pressure
and temperature variations as well as for inter-species cross-sensitivities, which are related to the overlapping spectral absorp-
tion regions of different trace species (Hammer et al., 2013).

The second available analyzer was an Off-Axis Integrated Cavity Output Spectroscopy gas analyzer (OA-ICOS), namely the
Los Gatos Ultraportable Carbon Analyzer, from here on called ICOS-analyzer. The instrument is field portable (weight of 17
kg) with a potential to run on battery power, so that it could be used to measure fluxes at different field locations around the K34
tower. The instrument measures $CO_2$, $CH_4$, CO and $H_2O$ at a flow of 0.3 LPM. For this study, the ICOS-analyzer was only
used to measure mixing ratios and fluxes of $CO_2$: since the CO concentrations in a pristine tropical forest are generally low,
the mixing ratios fell outside the reliable measurement range of the ICOS-analyzer. For this reason, all reported CO mixing





ratios and fluxes are based on measurements from the FTIR-analyzer.

## 2.3  Soil flux chamber measurements

Two intensive campaigns were held in 2020/2021, encompassing 9-days during dry season (DS campaign, 28 September-7
October 2020), and 7-days during wet season (WS campaign, 11-18 May 2021). During both campaigns, a series of soil flux
chamber measurements were performed on the plateau and in the valley. A soil chamber was made from a 200 L large bucket
(non-transparent), and fitting soil collars were made from stainless steel (15 cm height, 56.5 cm diameter). A strip of closed-
pore foam was glued to the inner edge of the chamber, so that no air could pass between the chamber and the collar during
measurement. Two holes were made on each side of the chamber at around 50 cm height where a quick connect ¼ inch fitting
was installed, serving as the inlet and outlet of the chamber. On the inside of the chamber, a four-inlet vertical sampling tube
was placed so that the air sampled (flow rate of 0.3 LPM) was a mix from different heights in the head space (∼10 cm, ∼25
cm, ∼35 cm, and ∼50 cm) (Clough et al., 2020). The setup (chamber and tubing) was tested for internal gas emissions under
field conditions (high temperature and humidity). For CO, an internal emission of <0.014 nmol s$^{-1}$ was found; the reported
CO fluxes are not corrected for this small possible internal emission.

Five soil collars were installed on the plateau (∼50 m from the tower), and five soil collars were installed in the valley
(∼50 m from the location of the nighttime valley measurements and valley stream (see section 2.5)), approximately one month
before the first (DS) measurement campaign. The valley soil collars were just far enough from the valley stream to not be
inundated after some days of heavy rain. The litter layer was not removed from the soil in the collars so that the soil surface
was representative of the forest floor. During each campaign, each collar was measured 3 times. Each collar was measured for
∼35 minutes, during which the air was circulated through the chambers by the internal pump of the ICOS-analyzer, which
measured $CO_2$ simultaneously. Right after chamber closure, a bag sample was sampled from the chamber inlet by use of an
external pump (KNF, NMP 830 KNDC B). After that, a subsequent bag was sampled every 10 minutes (4 bags in total). Air
was stored in 5L inert foil sampling bags (Sigma-Aldrich), which were brought to the FTIR-analyzer and analyzed on the same
day. The soil CO flux (F$_{CO}$) was calculated as follows:

$$F_{CO} = \frac{\Delta[CO]}{\Delta t} * \frac{V}{A} \tag{1}$$

wherein $\frac{\Delta[CO]}{\Delta t}$ was calculated with linear regression over the CO mixing ratios of the consecutive 4 bags, and $\Delta[CO]$ con-
verted from mixing ratios (nmol mol$^{-1}$) to concentrations (nmol m$^{-3}$) by the ideal gas law (assuming a T$_{air}$ of 25 °C), V is
0.20 m$^3$, and A is 0.25 m$^2$. After each measurement, soil temperature T (measured with a manual sensor, type TP-101) and



soil volumetric water content (VWC) (AT SMT150) were measured around the collar 5 times, of which the median was taken.

## 2.4  Plateau tower CO mixing ratios and flux estimates

To determine CO mixing ratios at different heights in the canopy air, inlet lines of Synflex tubing (¼ inch) were installed at
the tower at 36 m, 15 m and 5 m height. Each inlet was equipped with a rain protection cap and a particle filter. Each line extended until the cabin, where it passed an air cooler (4°C) with several water traps, which prevented condensation droplets from entering the sampling manifold and instrument. After the water traps, the lines led to a sampling manifold, from which one single line entered the FTIR-analyzer. Calibration gases (gas 1 with 381.8 $\mu$mol $CO_2$ mol$^{-1}$, and 431.0 nmol CO mol$^{-1}$, and gas 2 with 501.6 $\mu$mol $CO_2$ mol$^{-1}$ and 256.7 nmol CO mol$^{-1}$) were available, and measured at least 3 times during each
campaign. During the campaign-periods, the FTIR-analyzer alternated measuring air from the 3 heights in an half hourly cycle (10 min per height), using a sampling flow of 1.2 LPM. Since the FTIR-analyzer has a large measurement cell (3.5 L) and a corresponding long turn overtime, only the last 2 minutes of each 10 minute measurement window were used.

During the dry season campaign, a leak was found in the tower 36 m inlet line, so that the data from this inlet between
28 September and 4 October could not be used. For this reason, for some of the subsequent analyses, measurements from an extended period were used, namely 5-12 October. The daytime tower vertical profile measurements were interrupted during campaign days because the instrument was used to measure the different sampling bags, sampled in the ecosystem. To estimate ecosystem CO fluxes from atmospheric CO mixing ratios, only nighttime CO measurements were used.

The measured CO mixing ratios were interpreted using 2 different approaches. Nighttime vertical CO mixing ratio profiles (dCO/dz) were compared between different time windows over each night. To enable a more straightforward comparison, the mixing ratios at 15 m and 36 m were expressed relative to the 5 m height (dCO-15m and dCO-36m): a negative dCO indicates that the CO mixing ratios at 15 m or 36 m are lower than at 5 m height. Vertical profiles per ∼1-h time window were calculated, which consisted of 3 measurements per height. Per night, the following time windows were used: 18h-19h, 20h-21h, 22h-23h,
0h-1h, 2h-3h, 4h-5h. The given date of a night indicates the date of the start of the evening, for example ´28 September´ indicates the night from 28-29 September. Please note that the ´d´ is used to indicate a spatial difference (vertical profile, dCO-36m), while the $\Delta$ symbol is used to indicate a change over time (introduced below).

Next to the analyses of the vertical CO profile, a canopy layer budget method was used, as described by Trumbore et al.
(1990) and applied by earlier studies in tropical forests for $CH_4$ (Carmo et al., 2006):

$$\frac{\Delta CO}{\Delta t} = PCO - k(C(CO) - C(COatm)) \tag{2}$$



wherein PCO stands for the production of CO in the canopy layer, C(CO) and C(CO$_{atm}$) stand for respectively the mixing
ratio in the canopy layer and the mixing ratio of the overlying atmosphere, and $k$ represents an exchange coefficient. This
equation can also be defined for $CO_2$, which can then be merged into:

$$\frac{\Delta CO}{\Delta CO_2} = \frac{PCO - k(C(CO) - C(COatm))}{PCO_2 - k(C(CO_2) - C(CO_2atm))} \tag{3}$$

During stable nighttime conditions, when the exchange between the canopy layer and the overlaying atmosphere is low, a
similarity between $CO_2$ and CO mixing ratio patterns and production rates can be assumed, so that Eq. 3 can be simplified to
(Carmo et al., 2006):

$$PCO = \frac{\Delta CO}{\Delta CO_2} * PCO_2 \tag{4}$$


in which PCO$_2$ can be inferred from Eddy Covariance flux data. To filter for nighttime stable conditions, the period 18h-4h
was chosen, based on an earlier study at this field site showing generally stable conditions for these hours (Araújo et al., 2002).
For each night of the campaign week, the $\frac{\Delta CO}{\Delta CO_2}$ was calculated for different time windows 18h-20h, 20h-22h, 22h-0h, 0h-2h,
2h-4h. The two heights below the canopy, namely 5 and 15 meters, were both used independently, and values shown are fil-
tered for $R^2$ >0.9. Due to unavailable micro-meteorological $CO_2$ flux measurements, it was decided to choose a fixed value
for PCO$_2$ of 7.8 $\mu$mol m$^{-2}$ s$^{-1}$, based on a previous study at the same field site (Chambers et al., 2004).

## 2.5 Valley CO mixing ratios and flux estimates

To complement the mixing ratio measurements on the plateau, additional measurements were performed in a valley close to the
K34 tower. Equipment was placed in a box on a wooden boardwalk, constructed above a stream and a muddy, and sometimes
inundated, area. Two 10 m ¼ inch teflon lines were extended from the Zarges box and installed ∼10 m from the boardwalk
(∼2 m from the valley stream), hanging 1 m above the soil surface. The Zarges box contained the ICOS-analyzer, which was
continuously sampling air from one teflon line (0.3 LPM, measurement every 10 sec). In addition, a sampling device with
a KNF pump (NMP 830 KNDC B, ∼1 LPM) was placed in the same box, continuously flushing the $2^{nd}$ sampling line. At
fixed times (+0h, +3h, +6h, +9h after start of the measurements), air (∼8 liters) was sampled into bags (4 bags, 10L inert foil,
Sigma-Aldrich). These bags were collected during the following morning, and measured by the FTIR-analyzer on the same





day. The starting time of the measurements was usually just around nightfall, between 17:30 and 18:30, and the external battery feeding the ICOS-analyzer usually held approximately 10-12 hours.

The continuous ICOS-analyzer measurements were used to study the general behaviour of the $CO_2$ mixing ratio trends during the night, while the additional bag measurements were used to determine the CO nighttime increase and the $\frac{\Delta CO}{\Delta CO_2}$ ratio (Eq. 4). For $PCO_2$, the value of 7.8 $\mu$mol m$^{-2}$ s$^{-1}$ was used (Chambers et al., 2004).

### 2.6 Laboratory thermal degradation measurements

To study thermal degradation of ecosystem material, a simple laboratory experiment was set up. Soil (upper 3 cm, not sieved) and senescent leaf material was sampled from a 2x2 m$^2$ area on the plateau and in the valley. The material was dried at 35 °C for 72 h, to assure microbial activity to be negligible (Lee et al., 2012). From each material, 3 sub samples were taken of ∼2 grams (leaves) and ∼30 grams (soil). For the experiment, a glass flask (inner diameter = 6.7 cm, height = 15 cm) was placed in a closed loop with the FTIR-analyzer. For this experiment, only glass and stainless steel material was used. Blank measure-
ments showed the set up was not emitting CO. The sample material was distributed equally in the flask. The samples were heated in temperature steps of 5°C (20–65 °C) by use of a controlled temperature water bath. Temperature time steps were 20 min. During the experiments, air was circulated between the glass flask and the FTIR-analyzer and measured once per minute.

The production rate of CO was derived from the measured mixing ratio change over time, and is expressed as nmol CO
gr$_{leaves}^{-1}$ min$^{-1}$ (or nmol CO gr$_{soil}^{-1}$ min$^{-1}$). To be able to express senescent leaf CO production rates on ecosystem scale, a literature senescent leaf density value of 117 and 67 g m$^{-2}$ (1.17 and 0.67 t DW ha$^{-1}$) was taken for respectively plateau and valley, as measured by Luizão et al. (2004) at the same field site. To be able to express soil material CO production rates from a 10-cm soil layer on ecosystem scale, a plateau soil bulk density of 1.05 g cm$^{-3}$ was assumed, as measured on the same field site (Marques et al., 2013). All experiments were conducted under dark conditions, to exclude photo-degradation fluxes.

## 3 Results

### 3.1 Soil CO and $CO_2$ fluxes

On the plateau, CO fluxes determined from the accumulation in the soil chambers were significantly larger in the dry season than in the wet season (Fig. 1, Table 1), and one collar in the wet season even showed uptake during all three measurements. When grouping all plateau measurements (dry and wet season), a correlation with soil temperature ($R^2$=0.53) and an inverse
correlation with soil moisture ($R^2$=0.57) was found (Fig. 2). Plateau soil moisture (VWC) and soil temperature (T) values showed a clear correlation, with higher T accompanied by lower VWC ($R^2$=0.70). Valley CO fluxes were generally higher than plateau CO fluxes. As on the plateau, valley wet season fluxes were smaller than dry season fluxes (Table 1), but only





a weak correlation with soil T ($R^2$=0.24) and soil VWC ($R^2$=0.13) was found. Also in the valley, warmer temperatures were accompanied with lower VWC, although the correlation was weak ($R^2$=0.15). For $CO_2$, dry season fluxes were higher in the
valley than on the plateau, while in the wet season the pattern was inverted. Nevertheless, differences in $CO_2$ fluxes between seasons and topographic locations were not significant (Table 1).

### 3.2 Atmospheric CO mixing ratios and ecosystem CO flux estimates

Dry season CO campaign mixing ratios varied between 127 and 292 ppb (Fig. 3). Mixing ratios between the different heights generally followed a common pattern, indicating that air masses with elevated CO passing the tower are also reaching lower
forest levels. Wet season campaign CO mixing ratios ranged between 94 ppb and 250 ppb, and generally showed less variation, fewer peaks, and lower mixing ratios. It is expected that (part of) the elevated mixing ratios and passing peaks in the dry season can be explained by the presence of biomass burning plumes, which can be transported over long distances (Andreae et al., 2012). The CO mixing ratio patterns, and the possible trajectories and dispersion of these biomass burning plumes, are subject of ongoing research and will not be further discussed in this study.


Vertical profiles per 1-h time windows are shown in Fig. 4. In the dry season, 4 out of 7 nights showed constant decreases in CO mixing ratio from 5 to 36m, i.e. consistently negative dCO-36m during the whole night. Average nighttime dCO-36m values for these nights were -10.5 ppb, -8.1 ppb, -10.9 ppb and -7.0 ppb (6, 7, 9 and 10 October, last column of Fig. 4). In the wet season, vertical nighttime profiles generally showed smaller variation in CO mixing ratios, however still 5 out of 7 nights
showed consistently decreasing CO mixing ratios with height over the whole night. Average dCO-36 values for these nights were generally smaller than in the dry season: -2.5 ppb, -5.6 ppb, -4.6 ppb and -2.3 ppb (resp. 13, 15, 16 and 17 May), with the exception of 14 May (-12.9 ppb) (Table 1, Fig. 4). Since no micro-meteorological measurements are available for the campaign periods, we cannot hypothesize why the night of 14 May was divergent.

The canopy layer budget method was applied to the plateau below-canopy inlet heights 5 and 15 m, and calculated $\frac{\Delta CO}{\Delta CO_2}$ ratios with an $R^2$ >0.9 were selected, which was 29% and 45% of the cases in the dry season, and 41% and 40% in the wet season for 5 m and 15 m respectively. Dry season $\frac{\Delta CO}{\Delta CO_2}$ ratios were slightly higher than wet season ratios, but differences were not significant (Table 1). Applying Eq. 4 to the 5 m mean nighttime ratios (DS=0.27 and WS= 0.24) gives a plateau ecosystem net production estimate of 2.1 and 1.9 nmol CO m$^{-2}$ s$^{-1}$ for the dry season and wet season respectively. The canopy layer
budget was also applied to nighttime valley measurements (inlet at ∼1 m height). In the dry season, correlation coefficients ($R^2$) between $\Delta CO$ and $\Delta CO_2$ reached 0.75 in 8 out of 9 nights (4 nights with $R^2$ >0.9), and in the wet season, 6 out of 7 nights reached $R^2$ >0.75 (5 nights with $R^2$ >0.9). The wet season ratios were significantly higher than the dry season ratios, and applying Eq. 4 to the mean ratios leads to estimates of a net valley CO production of 1.1 and 2.3 nmol m$^{-2}$ s$^{-1}$ for respectively the dry and the wet season (Table 1).





### 3.3 Laboratory results

Senescent leaves exposed to different temperatures emitted significant amounts of CO at rates increasing exponentially with higher temperatures. At 25 °C, average emission rates of 0.006 and 0.002 nmol CO $g_{leaves}^{-1}$ min$^{-1}$ were measured for plateau and valley samples respectively. The estimated ecosystem CO production rates, based on these average emission rates and literature senescent leaf density (Luizão et al., 2004), are 0.012 and 0.002 nmol CO m$^{-2}$ s$^{-1}$ at 25 °C for respectively the plateau and valley ecosystem (Fig. 5). The plateau soil material showed clear CO production increasing with higher temperatures. The valley soil material also showed CO production, but due to instrument problems, a complete and consistent data set could not be collected. For plateau soil material, an emission of 0.0005 nmol CO $g_{soil}^{-1}$ min$^{-1}$ at 25 °C was measured. The estimated ecosystem emission coming from a plateau soil layer of 10 cm (Marques et al., 2013) at 25 °C was estimated to be ~0.9 nmol CO m$^{-2}$ s$^{-1}$ (Fig. 5).

### 4 Discussion

Plateau and valley soil chambers, measuring the emission of soil and litter together, generally showed net emission of CO, except for one plateau collar in the wet season. On the plateau, CO fluxes showed a relation with soil temperature (R$^2$=0.53) as well as with soil VWC (R$^2$=0.57). In the valley, there was less variation in soil temperature and soil VWC, so that possible dependencies were less pronounced. Correlations between soil temperature and soil VWC (R$^2$=0.70) on the plateau did not permit determination of which factor is driving the soil CO flux variation.

Abiotic CO production as well as microbial CO uptake should correlate positively with increasing soil temperatures (Cowan et al., 2018; King, 2000; Lee et al., 2012; Derendorp et al., 2011; van Asperen et al., 2015; Moxley and Smith, 1998). While for microbial CO uptake the relationship is expected to have an optimum temperature (King, 2000), the abiotic thermal degradation fluxes are exponential (Derendorp et al., 2011; van Asperen et al., 2015; Lee et al., 2012; Conrad and Seiler, 1985) so that CO production is expected to become dominant at higher temperatures (Moxley and Smith, 1998; King, 2000; van Asperen et al., 2015; Cowan et al., 2018). The role of VWC is more complicated. On the one hand, lower soil VWC leads to higher soil diffusivity, enhancing the CO uptake, thereby shifting the balance to soil CO uptake. For example, at the same field site, high local CO uptake was observed from termite mounds, which consist of dry porous material (van Asperen et al., 2021; Martius et al., 1993). On the other hand, the availability of soil moisture has a direct effect on microbial CO uptake. Several studies found a parabolic response, with soil CO uptake having an optimum at VWC ~20-30% (Moxley and Smith, 1998; King, 2000; King and Hungria, 2002). Based on the supposed decrease in CO uptake when VWC >30%, one would expect a shift towards more positive CO fluxes from the dry season to the wet season, which is opposite of what is observed in our measurements (Fig. 1). Following this line of reasoning, we expect that the observed negative correlation between VWC and soil CO fluxes is an indirect one, driven by the correlation of soil T and soil VWC.



The laboratory experiment, isolating the effect of temperature on CO production of senescent leaves and soil material, indicated a clear exponential increase in CO emissions with temperature, as also reported by earlier studies (Derendorp et al., 2011; van Asperen et al., 2015; Lee et al., 2012). By combining literature values with our laboratory results, a simple approximate calculation was done to estimate the CO emission of senescent leaves and soil material at 25°C at the ecosystem scale. Senescent leaves in the amount expected in the surface litter layer (Luizão et al., 2004) were estimated to emit respectively 0.012 and 0.002 nmol CO m$^{-2}$ s$^{-1}$ on the plateau and in the valley, and a 10-cm plateau soil layer (Marques et al., 2013) was estimated to emit 0.93 nmol CO m$^{-2}$ s$^{-1}$. This simple up scaling ignores the collocated simultaneous soil CO uptake and, more importantly, this estimate ignores the CO production of the entire soil column below. Nevertheless, this simple 'back-of-the-envelope' calculation already shows the potential of mineral soil to be a strong emitter of CO, and suggests that the observed chamber fluxes, which were measured over soil and litter together, mainly reflect emissions from the soil.

The laboratory CO emissions, the chamber CO fluxes, and the nighttime ecosystem CO increase all demonstrate net production of CO by this ecosystem. All observations were performed in absence of solar radiation, so that a photochemically-induced CO production pathway, such as photodegradation of organic material or the oxidation of VOCs and hydrocarbons (Lee et al., 2012; Schade et al., 1999; Tarr et al., 1995; Szopa et al., 2021), is unlikely to have contributed to our fluxes. In addition, the thick canopy of these forests prevent much sunlight from penetrating into the lower canopy or reaching the forest floor. Besides thermal degradation, ozonolysis of unsaturated hydrocarbons can produce CO in absence of radiation (Röckmann et al., 1998). However, CO produced via ozonolysis would be associated with a strong enrichment in $\delta^{18}$O, which was not observed in additional experiments (see Appendix A). Therefore, we can exclude that ozonolysis plays a major role in our ecosystem. Following this line of reasoning, and supported by the clear observed relationships between temperature and CO fluxes (Figs. 2 and 5), we conclude that thermal degradation is likely the main driver of the CO production in this central Amazon tropical rain forest.

Plateau and valley CO mixing ratios were used to estimate ecosystem CO fluxes, which was done by studying the vertical CO gradient (only on the plateau), as well as by applying a canopy budget method (plateau and valley). Both approaches were only applied to nighttime measurements, when atmospheric conditions are generally more stable and with locally produced gases ́trapped ́ below the canopy, so that mixing ratio changes are more pronounced (Araújo et al., 2002). The nighttime vertical CO gradients generally were negative i.e. had higher mixing ratios closer to the forest floor in comparison to above-canopy mixing ratios (Fig. 4). The vertical gradient can be used to estimate an ecosystem flux, by assuming a fixed canopy flushing rate of 90% over a vertical column of 30 m, as measured at a similar field site close-by (Simon et al., 2005). Querino et al. (2011) applied the same method and assumptions to vertical CH$_4$ gradients, measured at the same tower, where it was shown to give comparable flux estimates to on-site Eddy Covariance CH$_4$ measurements. The vertical CO gradients suggest an ecosystem flux up to ~2 nmol CO m$^{-2}$ s$^{-1}$, with vertical profiles generally indicating larger emissions in the dry season (Table 1, last column). On the plateau, the canopy budget method showed no significant differences in $\frac{\Delta CO}{\Delta CO_2}$-ratios between the wet and the dry season and, based on the 5m inlet, plateau CO fluxes were estimated to range between 1.25 and 4.13 nmol CO m$^{-2}$ s$^{-1}$



in the dry season, and between 0.94 and 3.28 nmol CO m$^{-2}$ s$^{-1}$ in the wet season. The valley nighttime $\frac{\Delta CO}{\Delta CO_2}$-ratios were generally lower in the dry season, and valley CO fluxes were estimated to range between 0.3 and 1.9 nmol CO m$^{-2}$ s$^{-1}$ and between 1.8 and 3.1 nmol CO m$^{-2}$ s$^{-1}$ for respectively the dry and the wet season.


The vertical gradient approach as well as the canopy budget method, and their subsequent ecosystem estimates, are possibly affected by the varying background CO mixing ratios (Fig. 3). We attempted to adjust for these background variations by applying strict filters. As described above, for the ecosystem flux estimate based on dCO/dz, only nights when the dCO-36m was negative over the entire night were selected for up scaling. For the canopy budget method, which is based on the temporal

change ($\Delta$CO) below the canopy, only consistent $\Delta$CO changes with a strong correlation to $\Delta CO_2$ were selected (R$^2$ >0.9), so that CO variations caused by a change in background mixing ratios are removed. The fact that the filtered ratios are relatively constant between heights and nights gives us confidence that this approach indeed selects the CO mixing ratio trends which are caused by the local ecosystem. In addition, the valley CO and $CO_2$ mixing ratios are possibly affected by nighttime drainage from the plateau (Tóta et al., 2008; Araújo et al., 2008): Araújo et al. (2008) showed valley nighttime $CO_2$ pooling at the same

field site, with plateau $CO_2$ laterally transported below the canopy. If this pooling is happening for $CO_2$, it is not unlikely that other trace gases, such as CO, are also transported. We therefore expect that the valley $\frac{\Delta CO}{\Delta CO_2}$-ratios are not affected or, in case only $CO_2$ is pooling, are underestimated, which would mean that valley CO emissions would be even higher. Thus, possible pooling would not affect our prediction that valleys are a net CO emitter.

The used PCO$_2$ of 7.8 $\mu$mol m$^{-2}$ s$^{-1}$, as reported by Chambers et al. (2004), is a general ecosystem value and not specified for season or topographic location. Different studies at this field site have demonstrated differences in $CO_2$ fluxes between the plateau and the valley. For example, de Araújo (2009) performed Eddy Covariance measurements on the plateau and in the valley, and found that the valley PCO$_2$ was 2/3 of the plateau PCO$_2$ (7.2 vs 4.8 $\mu$mol m$^{-2}$ s$^{-1}$). Comparing plateau and valley, Chambers et al. (2004) found that soil respiration at this field site was even twice as high on the plateau, but pointed out

that the valley soil respiration fluxes are likely underestimated to an unknown degree. Zanchi et al. (2014) found an opposite pattern, with valley soil respiration being ~1.5 times higher than plateau soil respiration, which is similar as observed in our study in the dry season (Fig. 1). Since the degree and direction of soil and ecosystem $CO_2$ flux variation between seasons and topographies is unclear, a differentiation in PCO$_2$ could introduce additional uncertainties. For this reason, for this study, it was decided to use a fixed PCO$_2$ for all topographies and seasons.


An overview of the *direct* soil CO flux measurements and *indirect* ecosystem CO flux estimates is given in Table 1. A direct comparison of these values should be done with care. First of all, the soil flux measurements are performed during (warmer) daytime hours, while the ecosystem estimates are determined for cooler nighttime conditions, although temperature variations below the canopy in this ecosystem are generally small (<7 °C, Araújo et al. (2002)). Secondly, the flux chamber is measuring

soil and litter only, while the ecosystem estimates include all possible sources and sinks below the canopy. Most importantly, soil flux values are measured directly, while the ecosystem fluxes are indirect estimates. Nonetheless, for the dry season, the





plateau soil CO fluxes and the ecosystem CO flux estimates agree on the sign as well as on the magnitude of the CO fluxes. Moreover, the $\frac{\Delta CO}{\Delta CO_2}$ ratio of the plateau nighttime increase shows similar ratios as the soil fluxes (Fig. 1, Table 1). We therefore expect that, in the dry season, the plateau nighttime CO increase is mostly driven by soil emission. For the wet season, the

plateau soil CO fluxes were a lot smaller than the estimated ecosystem CO fluxes, and the flux chamber even showed uptake at one location. Because of the decrease in soil CO fluxes, the soil flux $\frac{\Delta CO}{\Delta CO_2}$ ratios strongly dropped, which was only weakly observed in the ecosystem $\frac{\Delta CO}{\Delta CO_2}$ ratios (Fig. 1). The difference in flux magnitudes and ratios indicates that, in the wet season, the type of soil surface, as measured in our soil chamber, is probably not the main driver of the plateau nighttime CO increase. Possibly, the plateau soil CO fluxes have large spatial variability, with our soil collars representing cooler/wetter spots than the

surrounding area. Additional (nighttime) soil CO flux chamber measurements would be crucial to verify this hypothesis.

All three methods on the plateau indicate higher CO emissions in the dry season although the difference is only significant for the direct soil flux measurements. Based on the earlier described relationship between soil temperature and soil CO flux (Fig. 2), and the clear increase of CO production with temperature (Fig. 5), we expect that the generally higher soil temper-

atures in the dry season cause the difference in CO fluxes between the seasons. In Appendix B, typical plateau dry and wet season soil temperatures from this field site are shown, which indicate that the general diurnal temperature pattern barely drops below the estimated 'soil-CO-uptake-threshold-temperature' of 25.2°C (Fig. 2), even at night or in the wet season (Fig. A1).

Just as on the plateau, the valley soil fluxes were significantly higher in the dry season. In the dry season, soil chamber CO

fluxes and ecosystem CO flux estimates agree quite well (Table 1), indicating that the nighttime valley CO increase is driven by sources such as our measured valley soil surface. Nevertheless, in the wet season, a clear discrepancy between the soil chamber flux magnitude and the valley ecosystem estimate was observed, indicating that the nighttime CO increase is driven by additional sources, that are not captured in the flux chambers. Since the wet season is characterized by frequent and high amounts of rain fall, the valley stream is frequently flooding its adjacent areas, which was also observed for the area below

the valley inlet. Streams and rivers are known to be sources of CO (Zuo and Jones, 1997; Campen et al., 2023; Valentine and Zepp, 1993), so that it is likely that our wet season nighttime measurements were dominated by the nearby valley stream and its inundated areas. On the whole, based on our observations, we expect that the valley is a net source of CO, with generally higher *soil* emissions in the dry season, but with likely higher overall *ecosystem* emissions in the wet season, due to the contribution of the valley stream and its inundated areas.


Our measured soil fluxes are generally higher compared to the limited previous soil CO flux studies performed in (sub-) tropical ecosystems. Kisselle et al. (2002) observed uptake of -0.31 to -0.07 nmol m$^{-2}$ s$^{-1}$ in the dry and wet season, respectively (Brazilian savanna, opaque chambers, not burned area). Sanhueza et al. (1994) found that Venezuelan grasslands were a net CO source of 0.6 nmol m$^{-2}$ s$^{-1}$, which turned into a small CO sink when ploughed ($\sim$-0.3 nmol m$^{-2}$ s$^{-1}$). Venezuelan

forest soils were found to be a net sink ($\sim$-4 nmol m$^{-2}$ s$^{-1}$), but a net source ($\sim$0.1 nmol m$^{-2}$ s$^{-1}$) after deforestation and ´conversion´ into a scrub grass savanna Scharffe et al. (1990). To our knowledge, only one study before attempted to estimate





tropical rain forest CO fluxes, which was done by Kirchhoff and Marinho (1990) in a field site close-by ZF2 (Ducke Forest reserve forest, ∼50 km). Kirchhoff and Marinho (1990) measured the vertical CO gradient below the canopy, and observed higher concentrations close to the soil surface in comparison to canopy height (dCO of -10 ppb), which is similar to gradients
observed in this study. Based on this gradient, they estimated a forest CO flux of ∼6 nmol m$^{-2}$ s$^{-1}$, implying the forest to be a source of CO.

By providing the first *direct* tropical rain forest soil CO flux measurements, and by complementing these observations with nighttime ecosystem mixing ratio measurements, we can confirm the hypothesis of Kirchhoff and Marinho (1990), and can
state that tropical rain forest ecosystems are likely a net source of CO. By a simple up scaling (averaging seasons and topographical locations of soil CO fluxes, Table 1), we derive an average tropical rain forest *soil* emission of ∼1 nmol CO m$^{-2}$ s$^{-1}$. Our nighttime measurements indicate that the general ecosystem CO emission might be higher than this value, since swampy/inundated areas and valley streams, which are abundant in these ecosystems, might represent local hot spots. Translating our soil (and ecosystem) CO flux estimate to a yearly value gives an estimate of 0.9 g CO m$^{-2}$ yr$^{-1}$ and, by assuming a
global tropical (evergreen) forest area of 17.8 x 10$^6$ km$^2$ (Liu et al., 2018), a total global tropical forest emission of ∼16.0 Tg CO yr$^{-1}$ is estimated.

By an innovate combination of methods and instruments, we were able to study the CO mixing ratios and fluxes over different temporal and spatial scales, even in a remote challenging ecosystem such as a tropical rain forest. In the absence of a
mobile CO analyzer, a more logistically challenging bag sampling design had to be employed, which is why the number of CO flux and mixing ratio measurements remains small, especially in the valley. By only focusing on two campaign weeks and two locations in a tropical rain forest, we realize that our study presents only a snapshot of the complex CO dynamics of a tropical rain forest. Nevertheless, our unique set of measurements shows that tropical forests are a net source of CO, likely dominated by soil CO emissions. In addition, in valley areas, river and water sources are expected to contribute to the overall ecosystem
CO emission. To further improve our understanding of the CO dynamics of a tropical rain forest, more in-situ tropical forest soil and ecosystem CO flux measurements are needed, focusing on the possible complex dependencies between CO fluxes, (soil) temperature and soil moisture, and other environmental variables. Moreover, the role of soil type (e.g. texture, organic matter layer, porosity) and the significance of streams and inundated areas should be investigated. With the recent availability of mobile CO analyzers, we anticipate more in depth studies, focusing on the different temporal and spatial scales of tropical
rain forest CO fluxes.

## 5  Conclusions

By providing the first direct CO flux measurements of tropical forest soils, we can show that, in this ecosystem, soil CO production generally dominates over soil CO uptake. Complementary measurements of nighttime CO mixing ratios also suggest



an overall net ecosystem CO emission, and estimated ecosystem CO fluxes were of the same sign and of similar magnitude as the measured soil CO fluxes. Thus, we can state that tropical rain forest ecosystems are likely a net source of CO, and we expect that soil emissions are the main contributor to the ecosystem CO emissions.

We observed that higher net soil CO emissions were accompanied by higher soil temperatures, and the warmer dry season
generally showed larger soil and ecosystem CO emissions. With an additional laboratory experiment, the effect of temperature on CO production of senescent leaves and soil material was studied. The results show the potential of the soil material to be a strong emitter of CO, and indicates that the observed chamber fluxes, which were measured over soil and litter together, are mainly driven by the soil. By excluding a large contribution of the process ozonolysis or a radiation-induced CO production pathway, we expect that the observed CO fluxes are mainly produced by the process of thermal degradation.

By a simple up scaling, we provide a first observation-based tropical rain forest soil emission estimate of $\sim$1 nmol CO m$^{-2}$ s$^{-1}$ (0.9 g CO m$^{-2}$ yr$^{-1}$), which leads to an estimated global tropical rain forest soil emission of $\sim$16.0 Tg CO yr$^{-1}$. Total ecosystem CO emissions might still be higher, since valley streams and inundated areas might represent local hot spots. To further improve these tropical forest ecosystem CO emission estimates, and to understand the complex dynamics between soil
uptake and emission and its dependencies on environmental variables, more in-situ tropical forest soil and ecosystem CO flux measurements are essential.

## Appendix A: CO production via ozonolysis

Ozonolysis is the process by which ozone (O$_3$) can initiate oxidation of unsaturated hydrocarbons via addition to the double bond. In subsequent reaction steps CO can be produced (Criegee, 1975; Paulson and Seinfeld, 1992). Ozonolysis can occur
in absence of radiation, and is therefore a potential contributor to our observed ecosystem nighttime CO increase. Ozone is known to be isotopically strongly enriched in $^{18}$O and $^{17}$O, with a typical $\delta^{18}$O value of 80 - 100 ‰. Röckmann et al. (1998) demonstrated that CO produced via ozonolysis inherits the strong $^{18}$O and $^{17}$O enrichment of O$_3$, because one of the O atoms is transferred from O$_3$ to CO. By determining the $\delta^{18}$O of the CO increase at night, the contribution of ozonolysis can be assessed.

To investigate a potential contribution of ozonolysis to the nighttime CO production, additional measurements were performed in September 2022 in the same valley where the nighttime valley CO increase was observed. A teflon line ($\sim$5 m) was placed at $\sim$2 m from the valley stream, at 1 m height. This line was sampled during four time windows: 17:00-17:30 (just before sunset), 21:00-21:30 and 21:30-22:00 (nighttime), and 7:30-8:00 (just after sunrise). Two pressurized air flasks (1L)
were sampled per time window, using a manual flask sampler (Heimann et al., 2022). To determine the isotopic composition of CO, these 8 flasks were sent to the isotope laboratory at the Institute of Marine and Atmospheric Research Utrecht (IMAU) of Utrecht University. Unfortunately, several flasks broke during transport, and only 3 flasks (1 night flask, and 2 morning flasks)





could be analyzed for CO and its isotopic composition (Table A1). A Keeling plot of these values results in an intercept of
-20.0 ‰. Even though only three samples could be analyzed for isotopic composition, and the increase in CO is only 20-30
ppb, the Keeling plot analyses show that the higher nighttime CO concentrations are not accompanied by an enrichment in
$\delta^{18}$O, which would be expected if the CO was produced by ozonolysis of unsaturated non-methane hydrocarbons (Röckmann
et al., 1998). Having excluded ozonolysis as a significant contributor, we attribute the nighttime CO production to thermal
degradation.

**Appendix B:  Soil CO flux as a function of soil temperature**

Continuous soil temperature measurements were not available for the campaign periods in 2020 and 2021. Fortunately, soil
temperature measurements were available for most of the year 2019. Soil temperatures at different depths were monitored in
10-minute intervals (STP01, Hykseflux). From previous measurements at the field site, it was observed that the plateau soil
temperature, measured with the manual sensor TP-101, agreed well with the continuous soil temperature measurements at 2
cm (differences generally <0.2 °C).

A simple soil temperature-based diurnal CO flux pattern was estimated, by use of the relationship shown in Fig. 2 ($F_{CO}$=$T_{soil}$*1.29
- 32.5), which indicates that soil CO uptake starts to dominate (the net flux turns from positive fo negative) when temperatures
drop below 25.2 °C. Fig. A1 shows the average soil temperatures of May 2019 and November 2019 (left y-axes), and the
calculated soil CO flux (right y-axes). These months were chosen because they were close to the campaign months of 2020
and 2021 (May and October), and because they presented an uninterrupted data set for the complete month.

While the *average* monthly temperatures did not drop below the threshold of 25.2 °C, individual nights sometimes showed
lower temperatures. The standard deviation of the average temperature (not shown) was used to estimate a CO flux range
(dotted lines), which shows that as well in the dry season as in the wet season soil CO *uptake* can take place. The daily
averaged dry season (November) and wet season (May) flux was resp. 1.22 and 0.63 nmol CO m$^{-2}$ s$^{-1}$, indicating that, based
on this simple model, the tropical forest soils are generally a CO source year round.

*Data availability.*  The measured CO and $CO_2$ mixing ratios and soil chamber fluxes, as presented in this study, have been uploaded to the
open-access repository of Zenodo and can be found at https://doi.org/10.5281/zenodo.10223554 (van Asperen, 2023). The soil temperature
data, as presented in the Appendix, are available on request from the co-authors AA, MS, PRT and JAFS.

*Author contributions.*  HA designed and performed the field experiment, and wrote the paper, AA, BF, and SF provided access to the logistics
and infrastructure of the INPA-LBA field site, LRO, TLX, JM assisted in the setting up of the research infrastructure before and after the



campaign weeks, SB assisted during the field campaign weeks and performed part of the flux measurements, MS, PRT, JAFS processed and provided the soil temperature data from the K34 tower, TR and CV analyzed and evaluated the isotopic flask samples, and TW, JN, TR, and
ST reviewed and commented on the manuscript.

*Competing interests.* The authors declare that they have no conflict of interest.

*Acknowledgements.* The study was funded by the DFG-project 'Methane fluxes from seasonally flooded forests in the Amazon basin' (project nr. 352322796). We are thankful for the support of the crew of the experimental field site ZF2, the research station managed by INPA-LBA (National Institute for Amazonian Research (INPA)- The Large Scale Biosphere-Atmosphere Research Program in the Amazon
(LBA)). We would like to thank Santiago Botía for providing the flasks for the ozonolysis experiment. We would also like to express our gratitude to the staff of LBA, for providing logistics, advice, and support during different phases of this research.





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





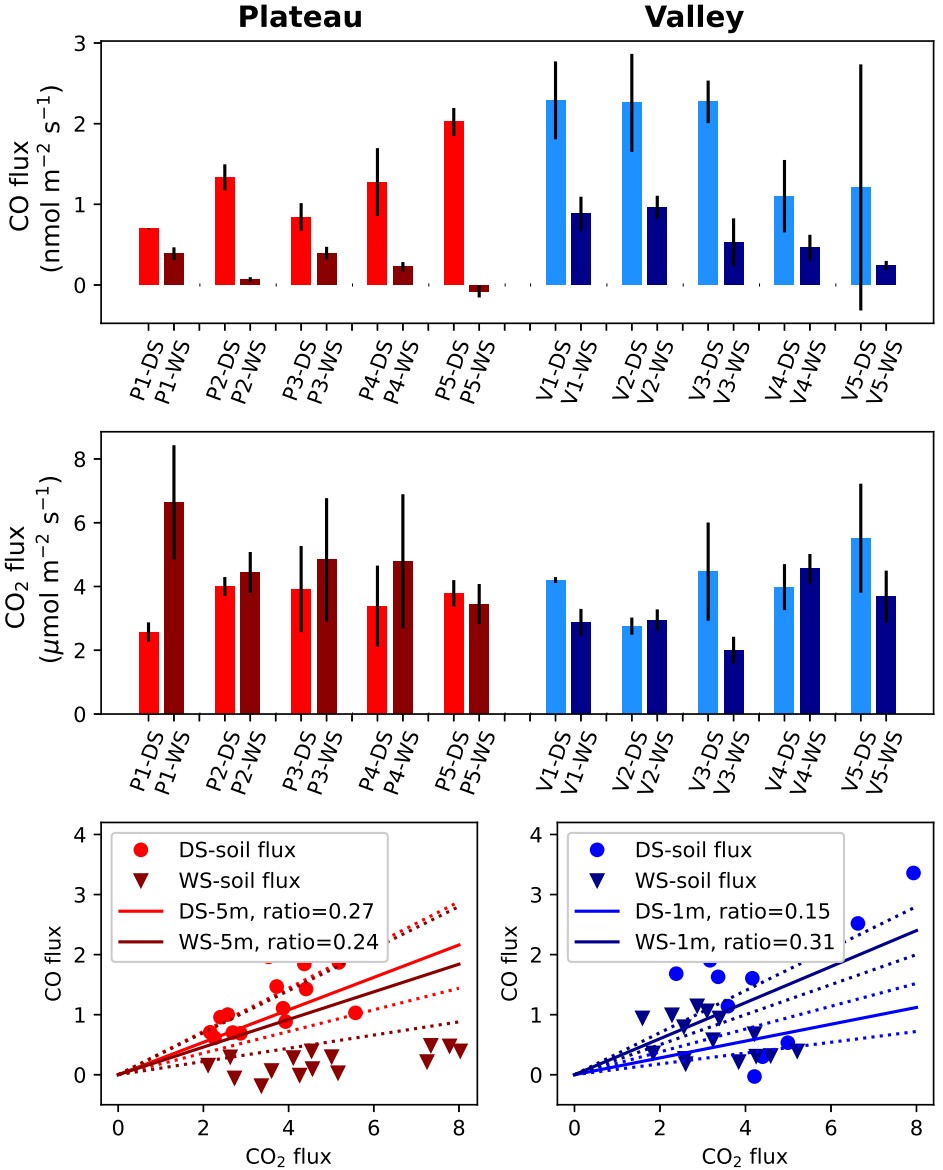

**Figure 1.** Upper and middle row; CO and $CO_2$ soil fluxes at 5 different locations on the plateau (left, reddish colors), and in the valley (right, blueish colors). Lighter colors indicate dry season (DS), darker colors indicate wet season (WS). Each location was measured 3 times during each campaign, the error bars indicate the standard deviation of the mean of the 3 measurements. CO fluxes are based on bag mixing ratio measurements, $CO_2$ fluxes are based on ICOS-analyzer measurements. The lower row shows the ratio between the soil CO and $CO_2$ fluxes (circles and triangles). In addition, the ratio's (and its standard deviation), as measured by the canopy layer budget method (tower height of 5 m), are shown in (dotted) lines.



**Figure 2.** CO fluxes (upper row) and $CO_2$ fluxes (middle row) on the plateau (left column) and in the valley (right column). Dry season measurements are indicated with circles, wet season measurements are indicated with triangles. Correlation coefficients between T (or VWC) and soil CO fluxes (or $CO_2$ fluxes) are given in the legend, and linear regression lines are shown for the cases where $R^2 > 0.5$. The formula given in the upper left graph indicates the relationship found between the manual soil temperature measurements and the measured CO fluxes. Lower row: correlation between T and VWC, correlation coefficient given in the legend.



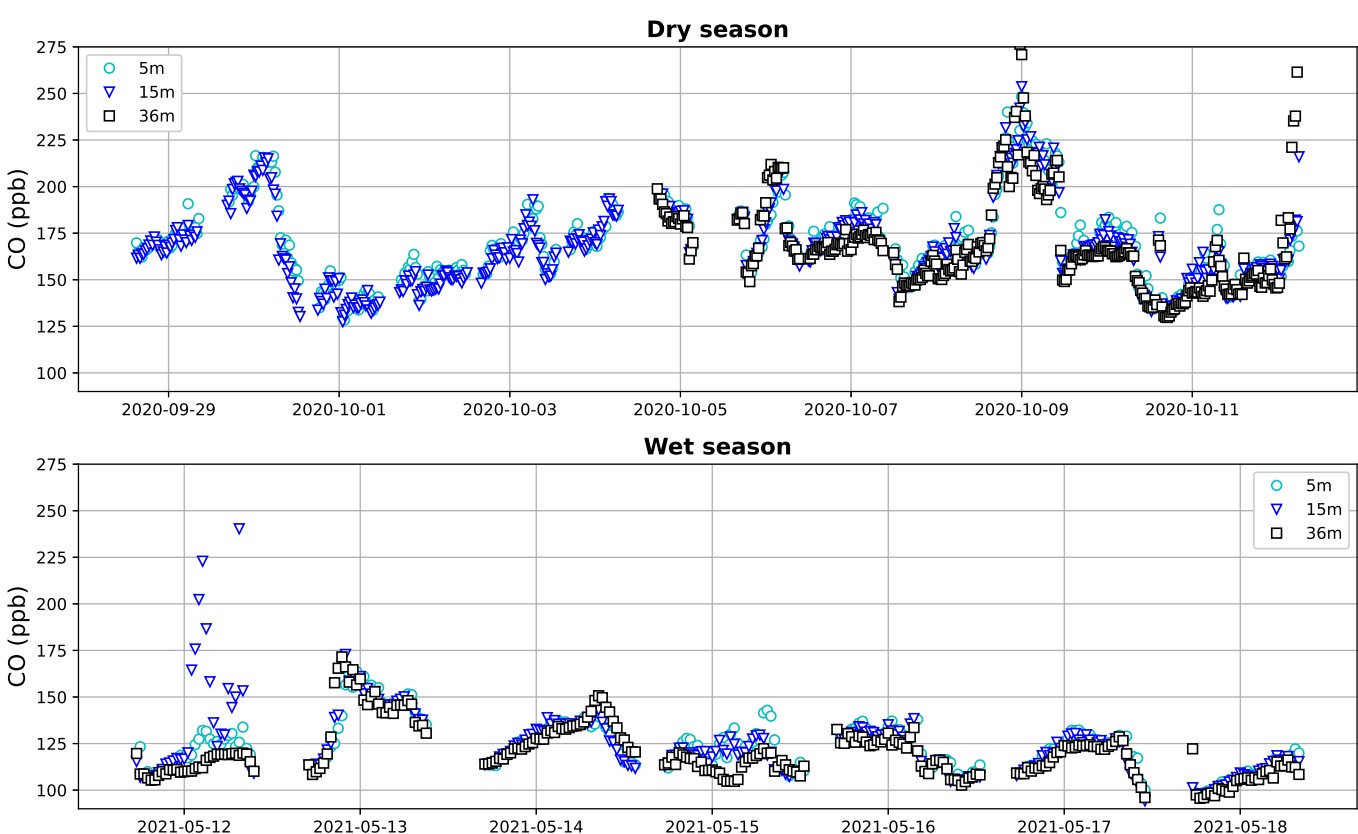

**Figure 3.** Tower CO mixing ratios during the dry season campaign (upper row) and the wet season campaign (lower row). Despite the variation, a general tendency, with higher CO mixing ratios below the canopy, is visible during both periods.



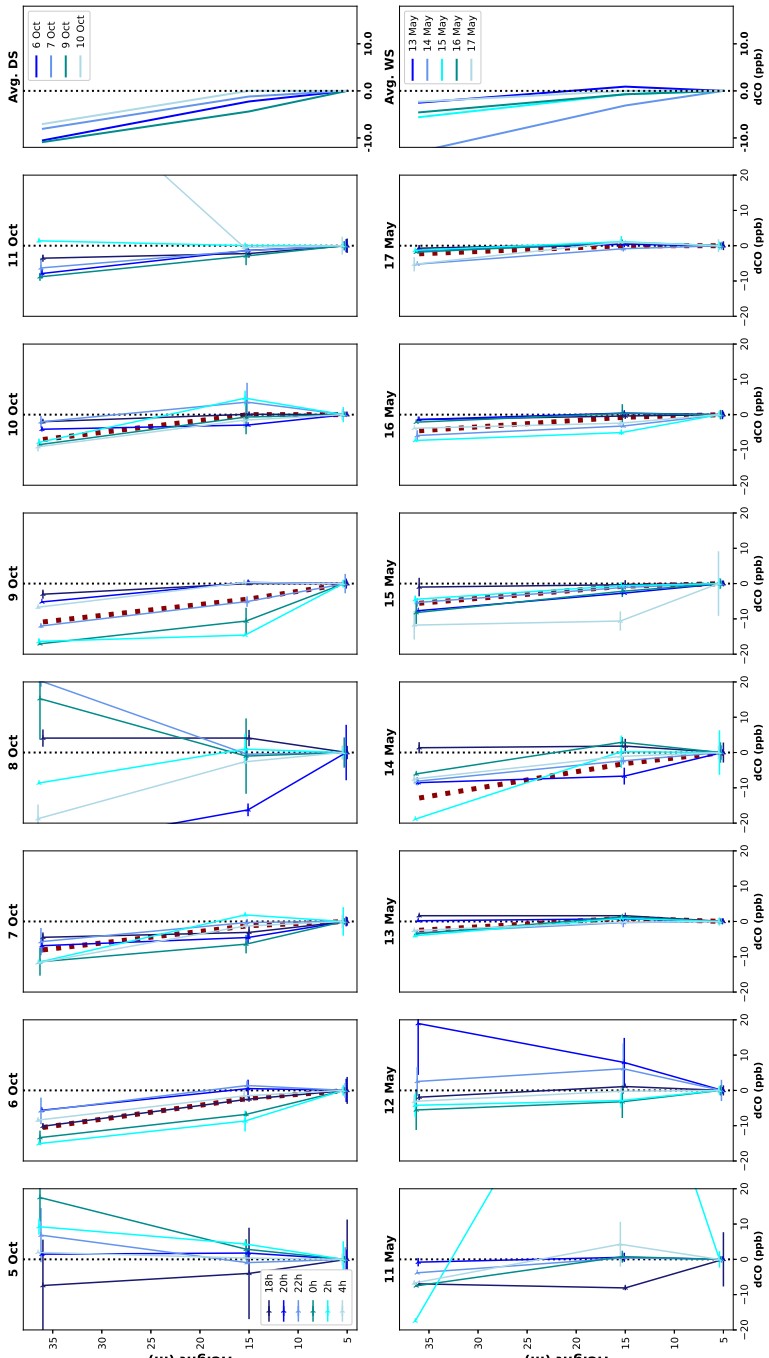

**Figure 4.** Vertical CO profiles for each dry season (DS, upper row) and wet season (WS, lower row) campaign night. The black dotted line indicates the zero line (dCO=0). An average nighttime CO profile was calculated if the 36 m height showed consistent lower CO mixing ratios in comparison to the 5 m height (dark red dotted line). The last column show these average CO profiles, and is plotted on a narrower x-axes range.





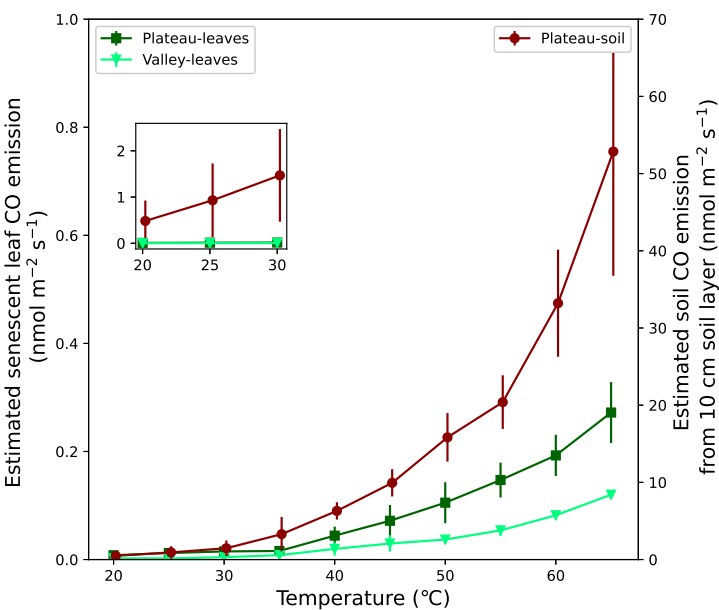

**Figure 5.** Expected CO emission of soil and senescent leaf material, expressed per surface area. For senescent leaf CO emission (left axes, green triangles and squares), the laboratory CO emissions (nmol $gr_{leaves}^{-1}$ min$^{-1}$) were converted to seconds (s$^{-1}$), and were multiplied by a senescent leaf density of resp. 117 and 67 $gr_{leaves}$ m$^{-2}$ for the plateau and valley ecosystem (Luizão et al., 2004), so that CO production is expressed in 'nmol m$^{-2}$ s$^{-1}$'. For the estimate of CO emission of a 10-cm soil layer (right axes, dark red diamonds), the laboratory emissions (nmol $gr_{soil}^{-1}$ min$^{-1}$) were converted to seconds (s$^{-1}$), and were combined with a soil bulk density of 1.05 g cm$^{-3}$ (Marques et al., 2013), so that soil CO production is expressed in 'nmol m$^{-2}$ s$^{-1}$'. The set-in figure shows a zoom-in of senescent leaves and soil emissions, plotted on the same y-axes scale, visualizing the expected dominance of soil CO emissions over senescent leaf CO emissions.





**Table 1.** Overview of the soil chamber CO and $CO_2$ fluxes and the different ecosystem CO flux estimates. The two left columns show the soil CO and $CO_2$ fluxes (range and mean (std)) as measured with the flux chamber technique. The third column shows $\frac{\Delta CO}{\Delta CO_2}$ ratios of these flux chamber measurements (range and mean (std)). The fourth column shows the $\frac{\Delta CO}{\Delta CO_2}$ ratio of the nighttime increase (range and mean (std), on which the estimated ecosystem CO flux is based (fifth column, canopy layer budget estimate, range and mean (std)). The second-to-last column shows the average nighttime vertical dCO-36m gradient (mean (std)), and the last column the estimated ecosystem CO flux (mean (std)) based on these dCO-36m values. DS stands for dry season, and WS stands for wet season.

| | **Soil CO flux** (chamber) nmol m$^{-2}$ s$^{-1}$ | **Soil CO$_2$ flux** (chamber) $\mu$mol m$^{-2}$ s$^{-1}$ | **$\Delta$CO/$\Delta$CO$_2$** (chamber) (-) | **$\Delta$CO/$\Delta$CO$_2$** (nighttime increase) (-) | **Ecosystem CO flux** (based on $\Delta$CO/$\Delta$CO$_2$) nmol m$^{-2}$ s$^{-1}$ | **dCO-36m** (vertical profile) nmol | **Ecosystem CO flux** (based on dCO-36m) nmol m$^{-2}$ s$^{-1}$ |
|---|---|---|---|---|---|---|---|
| Plateau DS | 0.62 to 2.26 1.23 (0.52) | 2.16 to 5.57 3.53 (1.02) | 0.18 to 0.65 0.35 (0.12) | 5m: 0.16 to 0.53, 0.27 (0.09) 15m: -0.15 to 0.53, 0.23 (0.12) | 5m: 1.25 to 4.13, 2.11 (0.70) 15m: -1.17 to 4.13, 1.79 (0.94) | -9.1 (1.9) | 1.4 (0.3) |
| Plateau WS | -0.18 to 0.49 0.20 (0.19) | 2.11 to 8.03 4.83 (1.87) | -0.05 - 0.11 0.04 (0.04) | 5m: 0.12 to 0.42, 0.24 (0.08) 15m: -0.69 to 0.54, 0.20 (0.25) | 5m: 0.94 to 3.28, 1.87 (0.62) 15m: -5.38 to 4.21, 1.56 (1.95) | -5.6 (4.3) | 0.9 (0.6) |
| Valley DS | -0.03 to 3.36 1.83 (0.97) | 2.38 to 7.93 4.18 (1.40) | 0.00 to 1.07 0.47 (0.27) | 1m: 0.04 to 0.25, 0.14 (0.25) | 1m: 0.31 to 1.95, 1.09 (1.95) | | |
| Valley WS | 0.19 to 1.14 0.62 (0.33) | 1.59 to 5.21 3.21 (1.00) | 0.06 to 0.59 0.22 (0.16) | 1m: 0.23 to 0.40, 0.30 (0.05) | 1m: 1.79 to 3.12, 2.34 (0.39) | | |

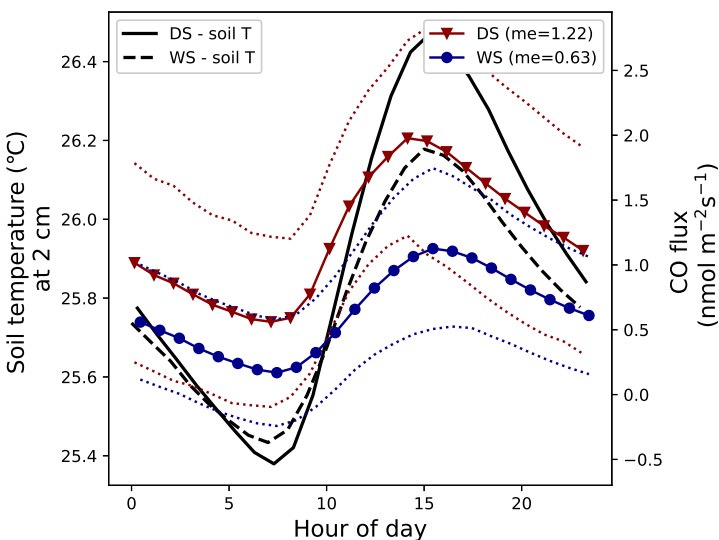

**Figure A1.** Left y-axes: Average soil temperature at 2 cm in the dry season (DS) of 2019 (October) and the wet season (WS) of 2019 (May) (standard deviations of the average soil temperatures are not shown). Right y-axes: Modeled soil $CO$ flux for the DS and WS (solid lines) and its standard deviations (dotted lines), based on the soil temperatures (and their standard deviation) at 2 cm depth (relationship shown in Fig. 2). The monthly mean calculated CO flux for the DS and WS is given in the legend, and is in nmol CO m$^{-2}$ s$^{-1}$.



**Table A1.** Sampling time of flask, measured CO mixing ratios (with sd in brackets), and $\delta^{18}$O-CO of flasks (with sd in brackets).

| Hour of sampling | Flask CO (ppb) | Flask $\delta^{18}$O-CO (‰) |
|---|---|---|
| 21:30-22:00 | 241.7 (0.5) | 13.3 (0.1) |
| 7:30-8:00 - first flask | 215.7 (0.2) | 17.3 (0.1) |
| 7:30-8:00 - second flask | 226.1 (0.3) | 16.4 (0.1) |
| Keeling plot intercept | | -20.0 (7.7) |