# Peer review of "The emission of CO from tropical rainforest soils"

_EGUsphere, 2023_

## Author Response (AR1)

**Answer to reviewer 1 (Jörg Matschullat, TU Bergakademie Freiberg)**

*14 April 2024, Hella van Asperen*

**Author:**

We would like to thank the reviewer for the time spent on the review and for the useful tips and comments. We appreciate the feedback and ideas on how to improve the manuscript! Here below you can find a point to point response to each of the comments and concerns.

*Reviewer:*

*The bottom line here is that the authors present an important and relatively little-studied topic in a clearly-written and well-presented manner. I recommend acceptance after very minor, mostly formal corrections.*

*Soil and ecosystem carbon monoxide fluxes were studied in both dry and rainy seasons as well as during daytime and nighttime at an established forest research site under the auspices of the National Institute for Amazonia Studies (INPA) in Manaus, Amazonas state, Brazil. The work encompasses complementary field site ground- as well as tower-based studies and additional laboratory measurements. The obtained results, albeit still a few only given the total "n" of data, fill a knowledge gap – and corroborate earlier hypothetical results from others. The inherent shortcomings are directly addressed within the manuscript, thus circumventing any doubt on the reader's side that individual results may have received undue emphasis.*

*The introduction very clearly sets the stage and inform readers about the state of art and understanding. Materials and methods clearly describe the methodology applied and related boundary conditions. However, in section 2.2 (as of line 100), I miss the distinct mentioning of Lower Limit of Detection and Determination, respectively, for both FTIR and ICOS analyzers.*

**Author:**

Thank you for your kind words and analyses. Concerning the missing of the 'Lower Limit of Detection and Determination', this is a good point which should be addressed.

The FTIR-analyzers CO precision ($\delta$) for 2 min spectral measurement is 0.45 ppb, and the $CO_2$ precision for 2 min spectral measurements is 0.05 ppm (van Asperen et al. 2015, Griffith et al. 2012). For the different applied methods, this results in the following detection limits:

- **Vertical CO gradient:** The detection limit of the '*Vertical CO gradient-ecosystem estimate*' can be defined as dCO-36m (the CO concentration-difference between 36m and 5m) being > $2\delta$ (>0.9 ppb CO). A dCO-36m of 0.9 ppb represents an estimated ecosystem CO flux of 0.15 nmol CO $m^{-2}$ $s^{-1}$. So, the vertical CO gradient method has a detection limit of 0.15 nmol CO $m^{-2}$ $s^{-1}$.

- **The canopy layer budget method:** This method is dependent on the FTIR's precision ($\delta$) for CO (0.45 ppb) as well as for $CO_2$ (0.05 ppm).

  - At the plateau, 2h-time windows were used. The average nighttime 2h-increase in $CO_2$ at 5m is 16.8 ppm (dataset available at Zenodo), so that the general nighttime ecosystem $CO_2$ buildup largely exceeds the FTIR's $CO_2$ precision ($2\delta$=0.1 ppm). Assuming the minimal-required dCO of 0.9 ppb ($2\delta$) over 2 h, an average $dCO_2$ of 16.8 ppm, and a $PCO_2$ of 7.8 µmol $m^{-2}$ $s^{-1}$, one would estimate a minimal detectable PCO flux of (0.9/16.8 x 7.8 ) 0.42 nmol CO $m^{-2}$ $s^{-1}$. The fluxes estimated in this study by this method for the plateau are higher than this detection limit.

  - In the valley, bags were sampled every 3 hours. The average 3h-increase in $CO_2$ at 1 m was ~50 ppm (dataset available at Zenodo), so that the general nighttime ecosystem $CO_2$ buildup largely exceeds the FTIR's $CO_2$ precision ($2\delta$=0.1 ppm). Assuming a minimal-required dCO of 0.9 ppb over 3 hours, an average $dCO_2$ of 50 ppm, and a $PCO_2$ of 7.8 µmol $m^{-2}$ $s^{-1}$, one would estimate a minimal detectable PCO flux of (0.9/50 x 7.8 ) 0.14 nmol CO $m^{-2}$ $s^{-1}$ . The fluxes estimated in this study by this method for the valley are higher than this detection limit.

- **The flux chamber method:** The chamber fluxes are calculated by use of the concentration difference over 4 bags. Requiring a minimum concentration difference of $2\delta$ (0.9 ppb=40 nmol $m^{-3}$) between four bags (30 min), the following minimum detectable flux can be calculated.

  FCO=dCO (nmol $m^{-3}$) / dt (10 min=600 sec) * V ($m^3$) /A ($m^2$)

FCO= 40/1800 * 0.200/0.25 → 0.01 nmol CO m$^{-2}$ s$^{-1}$.

For the manuscript, we suggest to add the following lines:

*Line 107: The precision (δ) of the FTIR-analyzers CO and $CO_2$ measurements for 2 min- spectral measurements is 0.45 nmol mol$^{-1}$ and 0.05 µmol mol$^{-1}$ respectively (van Asperen, 2015, Griffith et al. 2012). For the different methodologies based on concentration differences (explained below), a minimum concentration difference of 2δ was set as a detection limit.*

*Line 145: Requiring a minimum concentration difference of 2δ (FTIR CO δ=0.9 ppb) between the first and the last sampled bag, the minimal detectable flux of this system is 0.01 nmol CO m$^{-2}$ s$^{-1}$.*

*Line 253: In the wet season, vertical nighttime profiles generally showed smaller variation in CO mixing ratios, however still 5 out of 7 nights showed consistently decreasing CO mixing ratios with height over the whole night.* ***In addition, only three 2h-time windows showed a dCO-36m value < 0.9 ppb (2δ), which is considered the detection limit of the method.***

*Line 260: The canopy layer budget method was applied to the plateau below-canopy inlet heights 5 and 15 m.* ***All ΔCO$_2$ and ΔCO values of the 2h-window (plateau) and the 3h-window (valley) were higher than the set detection limit of 2δ (ΔCO$_2$>0.1 ppm, ΔCO>0.9 ppb).*** *Calculated ΔCO/ΔCO$_2$ ratios with an $R^2$ >0.9 were selected, which was 29% and 45% of the cases in the dry season, and 41% and 40% in the wet ΔCO season for 5 m and 15 m respectively.*

References:

- van Asperen, Hella. *Biosphere-Atmosphere Gas Exchange Measurements using Fourier Transform Infrared Spectrometry*. Thesis. Universität Bremen, 2015.

- Griffith, D. W. T., et al. "A Fourier transform infrared trace gas and isotope analyser for atmospheric applications." *Atmospheric Measurement Techniques* 5.10 (2012): 2481-2498.

***Reviewer:***

*In the results section, part 3.1, first paragraph (Lines 231 to 241), the title claims presentation of carbon monoxide and of carbon dioxide. However, only carbon monoxide is being addressed in the text. I suggest including CO$_2$ here. Generally speaking, all results are presented properly and clear.*

**Author:**

Thank you for picking up on this. We will include some sentences to also include the CO$_2$ flux results. The following sentence is proposed:

***Old sentence at line 239:*** *For CO$_2$, dry season fluxes were higher in the valley than on the plateau, while in the wet season the pattern was inverted. Nevertheless, differences in CO$_2$ fluxes between seasons and topographic locations were not significant (Table 1).*

***New proposed sentence at line 239:*** *For CO$_2$, dry season fluxes were higher in the valley than on the plateau, while in the wet season the pattern was inverted. Plateau CO$_2$ fluxes showed a small positive relationship with soil VWC, which is opposite of what was observed for CO. Valley CO$_2$ fluxes showed a positive relationship with soil T and a negative relationship with soil VWC. Differences in CO$_2$ fluxes between seasons and topographic locations were not significant and the observed relationships between CO$_2$ fluxes and soil T and soil VWC were weak (Table 1, Fig. 2), and will not be further discussed in detail in this manuscript.*

*Reviewer:*

*Discussion is appropriate and fine, so are the conclusions. I also appreciate the annexed additional information; good that it does not get lost that way.*

*The reference list needs a little work; there are quite a few bibliographically incomplete quotes and some mispellings.*

*In the following, all minor points will be listed, referring to their line numbers:*

1. *No line, List of authors: Co-author 8, Martha de Oliveira Sá, might be listed as member of INPA only. Her position as PhD student appears irrelevant here*
2. *As of line 28: Quotes throughout the manuscript are irregularly listed. I suggest to homogenize the listing and show authors in alphabetical order*
3. *Line 40/41: Ingersoll et al. 1974 is listed twice*
4. *As of line 59: The quotes at the end of the sentence, starting with Constant et al. 2008, should not be set in double brackets, but like "(Constant et al. 2008; Cowan et al. 2018; Pihlatie et al. 2016)". This issue reoccurs.*
5. *Line 67: Following the colon after "uncertain even as to sign", the sentence does not continue with a capitalized letter – as is often the norm after colons.*
6. *Line 71: The term "central Amazon" should be replaced by "central Amazonia" or similar to avoid misunderstandings.*
7. *As of line 87: When dashes are being used to say "from to", it should be an m-dash, no n-dash. This is valid throughout.*
8. *Line 88: "tree heights of …" not "three heights"*
9. *Line 102: Second single bracket after the quote should be taken out.*
10. *Line 130: "Five soils collars … from the tower), another five collars …" (another instead of and) to improve understanding.*
11. *Line 131: Brackets around "see section 2.5" is not necessary.*
12. *Line 136: "… was sampled from the chamber inlet, using an external pump" would be better.*
13. *Line 151: "… extended to the cabin, where…", not "until the cabin". Speaking of which: That cabin has not been mentioned before, but should be or explained here.*
14. *Lines 160/161: The sentence "For this reason, for some … namely 5–12 October" is completely unclear to me. Please rephrase.*
15. *As of line 232: It should be made unmistakably clear that the text here refers to carbon monoxide. This is not self-explanatory, since the related figure also displays $CO_2$.*
16. *As of line 252: "from 5 to 36 m" – the unit should not be written directly next to the number, a space is needed. Valid throughout in that context.*
17. *As of line 243: The authors use the unit "ppb". For gases this should be at least "ppb $v$" to unmistakably refer to a gas. Valid throughout.*
18. *Line 357: Araújo (2009) must likely be 2008.*
19. *As of line 400: I miss clear mentioning of the Cerrado biome here. For uninformed readers, just to say "in (sub)tropical ecosystems could be slightly misleading.*
20. *Line 406: "… into a scrub grass savanna (Scharffe et al 1990)." False bracket usage.*
21. *Line 407: "… CO fluxes, namely Kirchhoff and Marinho…", instead of "which was done"*
22. *Line 419. Dissect very long sentence in two: "… 0.9 g $CO_2$ m$^{-2}$ yr$^{-1}$. By assuming …"*
23. *Line 422: the word "innovate" seems not to fit here, better "innovative"*
24. *Line 434: "in-depth" (hyphen)*
25. *Line 499: The last quote, "(van Asperen 2023)" is missing in the reference list.*

*References, apart from misspellings or mission superscript/subscript settings:*

- *Laasonen et al. incomplete*
- *Marques et al. incomplete*
- *Sanderson: incorrect, must be MET Tech Notes 36: 8 p. plus web page.*
- *Van Asperen 2023 Zenodo is missing*

*Figure 4: Way too small to be well-legible after printing.*

*Again, the bottom line is that this work deserves publication and attention. Whether the upscaled CO emission of hypothesized ca. 16 TG CO per year is really such an enormous contribution I doubt, given the other orders of magnitude. Yet, it is valuable to not this.*

**Author:**

Thank you for paying attention to all these details. We will correct the errors in the reference list and the other points indicated by the reviewer. Regarding Figure 4, as also indicated by reviewer number 2, the visibility is indeed too low. We propose to move the left columns (the individual night figures) to the Appendix, and only show the most right column.

We appreciate your kind appreciation of this work and thank you for the time spent on the review!

**Answer to reviewer 2**

*15 April 2024, Hella van Asperen*

**Author:**

We would like to thank the reviewer for the time spent on the review and for the useful tips and comments. We appreciate the feedback and ideas on how to improve the manuscript! Here below you can find a point to point response to each of the comments and concerns.

**Reviewer:**

General Comments:

The manuscript submitted by van Asperen *et al*. describes the results of field and laboratory studies designed to address the question of whether tropical rainforest soils, and tropical rainforest ecosystems overall, serve as net carbon monoxide (CO) sources or sinks. The manuscript is a product of a large collaboration (16 authors on the manuscript alone) and represents a major undertaking to generate a rare dataset. The authors note that this is the first such CO dataset for tropical rainforest. This study represents an important step forward in understanding the CO cycle and also highlights the difficulty that experienced researchers face in collecting empirical data for small and difficult-to-measure fluxes against a backdrop of high environmental variability. The study also incorporates strong exploration of different potential sources of CO (ozonolysis, photodegradation, thermal degradation). The important direct measurements made in the field and under laboratory conditions are paired with modeling estimates and upscaling discussion in order to provide the scientific community with a sense of the magnitude and direction of CO fluxes from tropical rainforests. As such, the manuscript is a significant contribution to the CO literature and should be formally published. Below are suggestions and questions of clarification which might be useful to further improve the presentation of this work, but the manuscript does not, in my opinion, require further review.

Specific Comments:

One key constraint of this study is the absence of meteorological data for the time periods of the study. The authors point out that the data are not available; therefore, estimates from previous measurements are used to complete the analyses. These built-in assumptions and lack of data will necessarily limit the strength of the conclusions.

The strong difference in fluxes between wet and dry seasons and the separation of groups of data points, especially in Figure 2, could suggest that there are different mechanisms operating in different parts of the year. Examination of correlations across wet and dry seasons might mask the mechanisms that are operating in each season.

**Author:**

Thank you for your analyses. It is indeed true that different mechanisms might be dominant during different parts of the year. We suggest to highlight this better in the discussion by adding a sentence at line 377 and making adaptions to Figure 2. An elaboration on this topic can also be found later in this document.

*New sentence line 377: The difference in flux magnitudes and ratios indicates that, in the wet season, the type of soil surface, as measured in our soil chamber, is probably not the main driver of the plateau nighttime CO increase. Possibly, the plateau soil CO fluxes have **a** large spatial variability, with our soil collars representing cooler/wetter spots than the surrounding area. **In addition, the seasonal shift in correlations between soil temperature, soil VWC, and plateau soil CO fluxes (Fig. 2, left column) suggests that different dominant drivers may come into play during the wet season. A more elaborate flux chamber campaign, with possible nighttime measurements, would be crucial to verify the different hypotheses above.***

**Reviewer:**

Table 1 contains a great summary of the flux measurements and estimated fluxes based on vertical profiles. One way to examine these results is with a focus on building the ecosystem flux estimates from the multiple measured and estimated components, as one would read the table from left to right as it currently stands. However, to examine variability (in some cases, strong variability) in fluxes across sites or season, it would be far easier to read the table if the columns and rows were transposed so that a parameter could be compared across plateau and valley, wet and dry seasons. Another possibility is to separate the data into two tables by separating the chamber measurements from the profile measurements.

**Author:**

Thank you for your suggestions. We like your idea to transpose the columns and rows, so that comparison between plateau and valley is easier. We will prepare this table for the final version of the manuscript.

**Reviewer:**

Finally, while I agree that thermal degradation is likely to be an important factor determining overall CO production, I think that the rate of CO uptake, particularly in the litter layer, may also be a key driver of net fluxes. If the laboratory experiments involving senesced leaves and soil were all conducted under dry conditions with oven-dried materials, microbial uptake of CO would be suppressed. The manuscript acknowledges this simplification (line 308). Perhaps the shift to lower rates of net emission in the wet season is driven by high rates of CO uptake in the litter layer.

**Author:**

We agree with the reviewer that the role of CO uptake is important and should not be ignored in the simple upscaling. While acknowledged in line 308, we propose to elaborate on it, and would like to add the following sentence to line 308:

*Line 308: This simple up scaling ignores the collocated simultaneous soil CO uptake and, more importantly, this estimate ignores the CO production of the entire soil column below.* **Soil CO uptake has been shown to be dominant in ecosystems under specific conditions, such as lower temperatures and porous conditions (King 2000, Kisselle et al., 2002), and therefore in-depth research for this specific ecosystem would be needed to improve this upscaling.** *Nevertheless, this simple 'back-of-the-envelope' calculation already shows the potential of mineral soil to be a strong emitter of CO, and suggests that the observed chamber fluxes, which were measured over soil and litter together, mainly reflect emissions from the soil.*

Technical Comments:

Below are some more detailed comments, identified by line numbers.

Line 3 and elsewhere: "rainforest" is typically written as one word?

**Author:** we will make this consistent through the manuscript

Lines 27, 32, 246, and elsewhere: Are the parenthetical words essential or unessential modifiers?

**Author:** we will re-evaluate the use of the parenthetical words.

*- Line 27: Anthropogenic activities, such as  combustion of fossil fuel and biomass, contribute strongly to global CO emissions,...*

*- Line 32: Besides direct anthropogenic emissions, CO is also produced by atmospheric oxidation sources, such as the in-situ oxidation of methane and hydrocarbons, or can be emitted by (partly) natural sources such as forest fires, ocean emissions, the degradation....*

We prefer to leave this parenthetical word, since forest fires are not always considered as a pure natural source.

*- Line 246: It is expected that (part of) the elevated mixing ratios and passing peaks in the dry season can be explained by the presence of biomass burning plumes...*

We prefer to leave the parenthetical word, since we cannot exclude the contribution of other sources (than biomass burning) to the elevated mixing ratios and passing peaks.

Line 29: Comma after "large"

Line 38: "At the ecosystem level" or "On the ecosystem level"

Line 88: "tree" instead of "three"

Line 117 + (Section 2.3): The local meteorological conditions during the campaigns would be really useful to know, even if from field notes or a separate station. With such deep analysis of intensive field campaigns, the conditions at that time can be critical to interpretation. Are there are any data for air temperature and humidity? Are the data for water vapor available from the Los Gatos instrument? (line 112) Does the humidity vary across seasons and on a diel basis?

**Author:**

The Los Gatos instrument provides air humidity and temperature measurements, but experience learns that these measurements are somewhat dependent on instrument temperature (which is dependent on different variables such as time after switching on, local temperature, and whether the ventilation-fan is completely uncovered).

As discussed in the manuscript, the meteorological measurements were not completely functioning during the campaigns. No Eddy Covariance flux data are available for both campaigns. ZF2 tower (K34 tower) air temperature (51m) measurements were not running during the wet season campaign, but fortunately are available for the dry season campaign (from 29 Sep 2020).

We have looked for a local station to complement these measurements. The closest station (~50 km) with available measurements for these time periods is the local airport of Manaus (MAO), for which data can be found here:

https://mapas.inmet.gov.br/

In Figure X1 below, we have plotted the air temperature data from the MAO-airport for the 9-day period 28 Sep- 8 Oct (Dry Season, red dots), and for the 7-day period 11-18 May 2021 (Wet Season, blue dots). (Note: the wet season campaign was only 7 days, but for better comparison to the 9-day-Dry-Season campaign, also a 9-day period for the wet season is shown).

In the figure the available Dry Season- temperature data from the field site ZF2 is shown (black line). It is visible that the MAO-airport station is not completely comparable to fieldsite ZF2 since the airport is a not-forested area with higher daily maxima and lower daily minima. Nevertheless, from this figure it is clear that the '9-day-Dry-Season campaign' period generally had higher temperatures than the '9-day-Wet-season campaign' period. Even more pronounced are the differences for the relative humidity, which clearly shows lower daily values in the dry season compared to the wet season.

To not extend the amount of figures of the current manuscript, we have decided to not add this figure to the article. Nevertheless, we propose to add the text to the line 383:

*Line 383: Based on the earlier described relationship between soil temperature and soil CO flux (Fig. 2), and the clear increase of CO production with temperature (Fig. 5), we expect that the generally higher soil temperatures in the dry season cause the difference in CO fluxes between the seasons. **While no meteorological data from the K34-tower are available for both campaign periods, measurements from the local airport (~50 km distance) show that the dry season campaign period had clearly higher temperatures and lower relative humidities in comparison to the wet season campaign period (Inmet, 2024). In addition,** in Appendix B, typical plateau dry and wet season soil temperatures from this field site are shown (for 2019), which indicate that the general diurnal soil temperature pattern barely drops below the estimated 'soil-CO-uptake-threshold-temperature' of 25.2° C (Fig. 2), even at night or in the wet season (Fig. A1).*

Reference:

Inmet, 2024: https://mapas.inmet.gov.br/, Instituto Nacional de Meteorologia, station 8233-Eduardo Gomes, accessed on 15 April 2024.

[Figure]

Fig X1: Temperature (upper bar) and relative humity RH (lower bar) for the 9-day period of the Dry Season campaign (red) and a 9-day period around the Wet Season campaign (blue), as measured at the airport of Manaus (MAO, Inmet, 2024). The black dots in the background show the Dry Season temperature and relative humidity as measured at 51m at the K34 tower (fieldsite ZF2): measurements were not available for the wet season campaign period.

Line 130: To what depth were the soil collars installed? How close was the nearest vegetation?

**Author:**

The soil collars were installed to a depth of 5 cm. As well in the valley as on the plateau, no large vegetation was present next to the collars: most vegetation in this ecosystem consists of larger trees and palms, which are usually 2-3 m apart. In between these, you find areas which mainly consists of litter and some young mini-trees. For the selection of a soil collar location, we looked for positions >1m from trees (due to roots) and without any young trees. To clarify this, we propose to add the following information to the manuscript.

*Line 130: Five soil collars were installed on the plateau (~ 50 m from the tower), and five soil collars were installed in the valley (~ 50 m from the location of the nighttime valley measurements and valley stream (see section 2.5)), approximately one month before the first (DS) measurement campaign.* **Soil collars were installed until a depth of 5cm, and were installed at >1m from larger trees and bushes, containing only soil and litter.**

Line 157: "long turnover time"

**Author:** we will rephrase it to:

*Since the FTIR-analyzer has a large measurement cell (3.5 L) and a corresponding **long e-folding time**, only the last 2 minutes of each 10 minute measurement window were used.*

Line 251 paragraph: Might the vertical profiles be strongly influenced by temperature gradients and degree of vertical mixing? Based on previous data, the hours expected to be stable were chosen, but the meteorological conditions can be very different from day to day and year to year.

**Author:**

We agree with the reviewer that vertical profiles are influenced by temperature gradients and mixing. Dependent on the meteorological conditions, nighttime stability can be weakened or disturbed. For example, a strong rain effect can cause sudden mixing, breaking the nighttime stability and enhance canopy flushing. In case of only studying the vertical CO profile, one would see a smaller CO gradient (due to enhanced canopy flushing), which would lead to an underestimation of the ecosystem CO flux. Nevertheless, the 2nd method, the canopy budget method, makes use of the '$CO_2$/dt to CO/dt'-ratio, which are both equally affected by stability. Since estimates from this method gave very similar estimated flux magnitudes as the 'vertical CO profile'-method, we believe that the averaged estimated canopy flushing time of Simon et al. 2005 is of the correct magnitude.

Line 302+: As mentioned previously, the laboratory experiment under dry conditions with oven-dried materials may yield a different conclusion from what might occur in situ. Dry litter may show very low fluxes, but it could be significantly different under moist conditions, especially given its high surface area. "upscaling" in line 308.

**Author:**

Thank you for focusing on this. Yes, dry litter may show very low fluxes, which might be different (higher or lower) if the litter was naturally moist. While we expect that drier litter is easier degradable (for thermal degradation processes), a drier soil might also enhance soil diffusivity, and therefore enhance soil uptake. We prefer not to speculate too much on this part, and mostly would like to highlight the relative CO-producing potential of soil material (in comparison to present leaf material), which leads to our hypothesis that soil, and not litter, is the main contributor to our measured flux chamber CO emissions. This is expressed in the sentence at line 309:

*Nevertheless, this simple 'back-of-the-envelope' calculation already shows the potential of mineral soil to be a strong emitter of CO, and suggests that the observed chamber fluxes, which were measured over soil and litter together, mainly reflect emissions from the soil.*

Line 389 paragraph: Could this pattern be related to a change in uptake as well?

**Author:**

As visible in Fig 2, lowest figure on the right, the valley soil was slightly warmer and drier in the dry season. With **higher temperature**, one could expect higher soil biological CO uptake. With **drier** soil conditions, we expect higher soil diffusivity, and therefore higher soil CO uptake as well. So, the warmer and drier soil conditions in the dry seasons would lead to higher soil CO uptake (or indirectly lower CO emissions), which is opposite of what is observed in the field. Therefore we expect that the positive effects of high T on CO production are more dominant than the effects of high T on CO uptake, as also explained in the paragraph (lines 287-300).

Line 395: The sources of CO from streams and rivers are generally tied to photochemical processes of organic matter degradation. The discussion in this manuscript suggests that photodegradation is not a significant source of CO due to high canopy cover and lack of direct solar radiation. The explanation here is inconsistent with other discussion of photodegradation sources.

**Author:**

We understand that this parts seems inconsistent with the rest of the discussion, and we understand the need to clarify it here and in the manuscript.

First of all, we do not exclude the occurrence of photodegradation fluxes in this ecosystem. Nevertheless, the measurements we describe in this study are all performed under dark conditions (nighttime or non-transparent chambers), wherefore the fluxes which we have measured and estimated are produced by non-photodegradation sources. An interesting follow up topic would be to quantify the day time CO fluxes, which could represent the sum of thermal degradation and possible photodegradation-induced CO fluxes.

Concerning water CO fluxes: based on the literature review focused on ocean and wetland CO production, CO production seems dominantly attributed to photodegradation, but dark CO production has been observed before in oceans (Xhie, 2005, Zhang et al 2008) as well as in tropical rivers (Müller et al. 2015). Given the limited research performed on tropical (high-temperature) river and wetland CO concentrations and fluxes, we expect that the role of thermal degradation- produced CO is possibly understudied and not well quantified.

Considering only photochemically-induced production of CO: the amount of radiation reaching the forest floor is limited but still present, especially in the valley, which represents lower vegetation (Chambers et al. 2004). It is therefore possible that CO is produced photochemically in the valley stream, induced by direct and by diffusive radiation. As an exploratory hypothesis: it might be that daytime-produced CO in the valley stream (sub-)surface layers is still being released at night. Nevertheless, as far as we are aware, the magnitudes of CO production, the possible CO production at subsurface layers and the velocity of the CO release to the atmosphere (diffusion velocity) have not been studied for rivers and streams.

To sum up, the CO emitted by the valley stream at night might represent thermal degradation-induced fluxes as well delayed outgassing of photodegradation-induced CO fluxes, or a sum of both. Given the low resolution of our valley nighttime measurements, we prefer not to speculate on this further. To clarify this topic in the manuscript, we propose the following changes:

*Old sentence: 395: Streams and rivers are known to be sources of CO (Zuo and Jones, 1997; Campen et al., 2023; Valentine and Zepp, 1993), so that it is likely that our wet season nighttime measurements were dominated by the nearby valley stream and its inundated areas.*

*New sentence 395: Streams and rivers are known to be sources of CO,* ***either produced by photo or by thermal degradation*** *(Zuo and Jones, 1997; Campen et al., 2023; Valentine and Zepp, 1993,* ***Zhang et al 2008, Müller 2015****), so that it is likely that our wet season nighttime measurements were dominated by the nearby valley stream and its inundated areas.* ***Based on our measurements alone, we do not speculate whether the stream CO fluxes are caused by nighttime thermal degradation CO fluxes, or represent a delayed outgassing of photogradation-produced  CO.***

*Line 321: Following this line of reasoning, and supported by the clear observed relationships between temperature and CO fluxes (Figs. 2 and 5), we conclude that thermal degradation is likely the main driver of the* ***observed*** *CO production in this central Amazon tropical rain forest.* ***The possible presence of additional photodegradation-induced CO production during daytime would lead to even higher net total CO fluxes, a process not yet studied for tropical rain forests.***

Additional literature:

26.	Xie, H., O. C. Zafiriou, T. P. Umile, and D. J. Kieber (2005), Biological consumption of carbon monoxide in Delaware Bay, NW Atlantic and Beaufort Sea, Mar. Ecol. Prog. Ser., 290, 1 – 14, doi:10.3354/meps29000

27.	Zhang 2008: Dark production of carbon monoxide (CO) from dissolved organic matter in the St. Lawrence estuarine system: Implication for the global coastal and blue water CO budgets

28.	Müller, 2015: *Water-atmosphere greenhouse gas exchange measurements using FTIR spectrometry*. Diss. Universität Bremen; https://media.suub.uni-bremen.de/handle/elib/932?locale=de

Line 420:  There is an important change in terminology here. Whereas "tropical rainforest" has been used through most of the manuscript, the global scale estimate is related to "tropical (evergreen) forest" and "global tropical forest." Please clarify what area is specifically referenced here.

**Author:**

To be able to perform a simple upscaling which would be clear the reader, we tried to highlight which terminology we followed when referring to the manuscript of Liu et al 2018. Liu et al. 2018 divides tropical forests into 3 categories: 'tropical evergreen forest', 'tropical forested wetland', and 'tropical deciduous forest' (Table 4 of Liu et al. 2018). We believe that our ecosystem fits best to the term 'tropical evergreen forest', and have therefore chosen to use this value for upscaling.

We understand that this is unclear, and propose to change the sentence as follows:

*Old sentence: Translating our soil (and ecosystem) CO flux estimate to a yearly value gives an estimate of 0.9 g CO m$^{-2}$ yr$^{-1}$ and, by assuming a global tropical (evergreen) forest area of 17.8 x 10$^6$ km$^2$ (Liu et al., 2018), a total global tropical forest emission of ~ 16.0 Tg Co yr$^{-1}$is estimated.*

*New sentence: Translating our soil (and ecosystem) CO flux estimate to a yearly value gives an estimate of 0.9 g CO m$^{-2}$ yr$^{-1}$ and, by assuming a tropical rainforest area of 17.8 x 10$^6$ km$^2$ ('tropical evergreen forest', Liu et al., 2018, Table 4), a total global tropical forest emission of ~ 16.0 Tg Co yr$^{-1}$is estimated.*

Line 423: "innovate" should be "innovative"? Or perhaps another word? This approach is innovative for this biome perhaps?

**Author:** It should indeed be 'innovative'. Thanks for observing this.

Line 485: "2 cm depth"?

**Author:** It should indeed be '2 cm depth'. Thanks for observing this.

Line 488: "of" instead of "fo"

**Author:** We will correct this.

Line 490 and throughout (e.g., Figure 5): "axes" is plural for "axis" In this sentence, reference is to y-axis

**Author:** We will correct this.

Line 495: awkward sentence structure - perhaps "which shows that soil CO uptake can occur in the wet season as well the dry season"

**Author:** We will correct this.

Line 496 and Figure 5: spell out "resp."

**Author:** We will spell this out.

Figure 1: last line – "the ratios … at 1 m AND 5 m… are shown in solid lines. Dotted lines indicate standard deviation."

**Author:** We will correct this.

Figure 2: As mentioned previously, plotting all data together to derive correlation coefficients tends to ignore the strong separation of wet vs. dry season measurements. This pattern is evident in the plateau data and particularly the VWC vs. Soil Temperature plot.

**Author:** Thank you for pointing this out.

The variation in temperature and moisture within the campaigns was quite small, so that possible relationships were hard to determine. For this reason, we decided to group both seasons together. We agree that this approach ignores the existing differences between the wet and the dry season. We propose to change the following in the manuscript and improve Figure 2 (a new figure and caption you can find below).

*Old sentence line 234: When grouping all plateau measurements (dry and wet season), a correlation with soil temperature (R$^2$ =0.53) and an inverse correlation with soil moisture (R$^2$ =0.57) was found (Fig. 2). Plateau soil moisture (VWC) and soil temperature (T) values showed a clear correlation, with higher T accompanied by lower VWC (R$^2$ =0.70). Valley CO fluxes were generally higher than plateau CO fluxes. As on the plateau, valley wet season fluxes were smaller than dry season fluxes (Table 1), but only a weak correlation with soil T (R$^2$ =0.24) and soil VWC (R$^2$ =0.13) was found. Also in the valley, warmer temperatures were accompanied with lower VWC, although the correlation was weak (R$^2$ =0.15). For CO$_2$, dry season fluxes were higher in the valley than on the plateau, while in the wet season the pattern was inverted. Nevertheless, differences in CO$_2$ fluxes between seasons and topographic locations were not significant (Table 1).*

*New sentence line 234:* ***Soil temperature and moisture variation within campaigns was small, so that correlations to CO and CO$_2$ fluxes were not pronounced.*** *When grouping all plateau measurements (dry and wet season), a correlation*

[revised manuscript text omitted]

The correlation coefficients are probably R² (superscript) values, rather than subscript?

**Author:** We will correct this, as seen in the new figure 2 above.

Figure 3: What was the height of the canopy in the area of these measurements. Lines 87-88 describe the range of canopy height. Should it be assumed that 36 m is above the canopy?

**Author:** We will add the canopy height to the figure and explain it better in the text.

It would be useful to repeat the explanation here for the reason that there are no data for 36 m in the early dry season.

**Author:** We will repeat this.

Figure 4: The figure is quite difficult to read in its current form. Perhaps show one representative calculated profile along with average profiles, and the others can be provided in an appendix? Please also explain the error bars

associated with each height and the May 11 anomalous 15 m measurement. If the mixing ratio was not consistently lower at 36 m compared to 5 m (which assumes a net soil source always), is there a disturbance or mixing event that is assumed to have occurred?

**Author:** We agree with the reviewer that the visibility is too low. We propose to move the left columns (the individual night figures) to the Appendix, and only show the most right column in the manuscript. In addition, we will explain the error bars which are shown for each height and elaborate on what is visible for the profile of 11 May. To check whether a local disturbance occurred on 11 May, we propose to check the vertical $CO_2$ profile of that same night. We will elaborate on this in the new Appendix.

Table 1: Please consider whether transposing this table might ease comparisons across sites and seasons and allow for better consideration of the drivers of CO fluxes. Please also use abbreviations consistently throughout (e.g., sd vs. std) and indicate, if possible, the number of samples that the standard deviation is based on. Where no estimates were determined, "n.d." can be used to indicate "not determined."

 **Author:**

We like your idea to transpose the columns and rows, so that comparison between plateau and valley is easier. We will prepare this table for the final version. In addition, we will check the abbreviations and indicate the numbers of samples and add 'n.d.' for 'not determined'

Table A1. Perhaps should be called B1 because it is associated with Appendix B. The standard deviation on these measurements can be confusing, since each is still just one sample (n=1), so the "sd" is a measurement error.

Thank you for the opportunity to comment on this manuscript.

**Author:**
Thank you for reviewing the manuscript and the suggestions which will improve this manuscript!